# Motor cortex gates distractor stimulus encoding in sensory cortex

**Zhaoran Zhang** [1] **& Edward Zagha** [1,2] ✉

Suppressing responses to distractor stimuli is a fundamental cognitive function, essential for performing goal-directed tasks. A common framework for the neuronal implementation of distractor suppression is the attenuation of distractor stimuli from early sensory to higher-order processing. However, details of the localization and mechanisms of attenuation are poorly understood. We trained mice to selectively respond to target stimuli in one whisker field and ignore distractor stimuli in the opposite whisker field. During expert task performance, optogenetic inhibition of whisker motor cortex increased the overall tendency to respond and the detection of distractor whisker stimuli. Within sensory cortex, optogenetic inhibition of whisker motor cortex enhanced the propagation of distractor stimuli into target-preferring neurons. Single unit analyses revealed that whisker motor cortex (wMC) decorrelates target and distractor stimulus encoding in target-preferring primary somatosensory cortex (S1) neurons, which likely improves selective target stimulus detection by downstream readers. Moreover, we observed proactive top-down modulation from wMC to S1, through the differential activation of putative excitatory and inhibitory neurons before stimulus onset. Overall, our studies support a contribution of motor cortex to sensory selection, in suppressing behavioral responses to distractor stimuli by gating distractor stimulus propagation within sensory cortex.

To achieve our goals, we must respond to task-relevant target stimuli and not respond to task-irrelevant distractors. The ability to suppress behavioral responses to distractor stimuli is a component of impulse control, which is a core cognitive process. Impairments in distractor response suppression underlie some of the behavioral and cognitive dysfunctions in neuropsychiatric disorders such as attention-deficit/hyperactivity disorder (ADHD), obsessive–compulsive disorder, and substance abuse disorder[1,2]. A useful framework for understanding how the brain selectively responds to external stimuli is the Treisman attenuation theory[3]. According to this theory, target and distractor stimuli enter a short-term sensory store yet distractor signals are filtered out, or attenuated, at some point along its feedforward propagation. Evidence for attenuation occurring within neocortex is supported by studies across species[4–7]. Despite its importance and

long history of research focus, the mechanisms of target-distractor attenuation are poorly understood.

In this study, we explore the functional and physiological contributions of motor cortex to sensory selection. Motor cortices are most traditionally associated with motor planning and execution[8–10]. However, due to their strong direct and indirect connections with sensory regions, motor cortices are well-positioned to influence sensory processing. Prominent frameworks for the roles of motor cortex in sensory processing include corollary discharge[11–16], spatial attention[17–21], active sensing[22–24], sensorimotor integration[25], and sensory perception[26]. According to these frameworks, motor cortex can increase or decrease sensory encoding according to behavioral state and goal-direction. And yet, the circuit mechanisms of these sensory processing functions are just starting to be understood, due to the

[1]Neuroscience Graduate Program, University of California Riverside, Riverside, CA 92521, USA. [2]Department of Psychology, University of California Riverside, Riverside, CA 92521, USA. ✉e-mail: edward.zagha@ucr.edu

difficulty of combining neuronal recordings with causal perturbations during quantitative task performance.

We study sensory selection in the mouse whisker system: mice learn to respond to stimuli in one whisker field (target) and ignore stimuli in the opposite whisker field (distractor, also considered in the literature as a "non-target"). Previously, we identified a steep attenuation of distractor encoding downstream of its primary somatosensory cortex (S1) receptive field[7]. S1 receives robust top-down inputs from multiple regions including the ipsilateral whisker region of motor cortex (wMC), through both direct and indirect pathways[27]. Stimulation and suppression studies have found robust yet varied impacts of wMC activity on S1, including through excitation[22,23,26,28–31], inhibition[32,33], and dis-inhibition[34]. The impacts of wMC on S1 in the context of sensory selection is unknown.

In this study we combine wMC suppression with S1 single unit recordings in mice performing a target-distractor sensory selection task. We find evidence for wMC suppression of behavioral responses to distractor stimuli, by preventing the propagation of distractor stimuli into target-preferring S1 neurons.

## Results

### Performance in a selective detection task

Mice were trained to respond to transient whisker deflections in one whisker field (target stimuli) and ignore identical deflections in the opposite whisker field (distractor stimuli) (Fig. 1a-c). Given the lateralization of the mouse whisker representation, this task establishes target-aligned and distractor-aligned cortices that are symmetric across hemispheres and contralateral to the site of stimulus delivery (Fig. 1a). Performance in this task was quantified based on behavioral responses on target trials (hit rate), distractor trials (false alarm rate), and catch trials (catch rate). Mice were considered expert in this task once they achieved a discrimination d prime ($d'$, separation of hit rate and false alarm rate) >1 for three consecutive days. Optogenetic perturbations and electrophysiological recordings were performed in expert mice while performing this selective detection task. Two stimulus amplitudes were applied in each session (equal for target and distractor trials): "large" amplitude stimuli near the saturation of the hit rate psychometric curve and "small" amplitude stimuli within the dynamic range. Unless otherwise indicated, our data analyses reference the large amplitude stimuli.

### Suppression of target-aligned wMC or distractor-aligned wMC increases behavioral responses

We tested the impacts of wMC optogenetic suppression on task performance. Focal suppression was achieved by optical activation of GABAergic interneurons in either target-aligned (contralateral to target whisker stimuli) or distractor-aligned (contralateral to distractor whisker stimuli) wMC (Fig. 1d, f). Suppression was initiated 200–500 ms before whisker stimulus onset (Supplementary Fig. 1a), and was robust and stable throughout the post-stimulus lockout window. Control trials (light-off) and wMC suppression trials (light-on) were randomly interleaved. Suppression of either target-aligned wMC or distractor-aligned wMC increased hit rates, false alarm rates, and catch rates (Fig. 1g) (target-aligned wMC suppression, $n = 45$ sessions, hit rate: light-off trials: $0.94 \pm 0.02$, light-on trials: $0.96 \pm 0.02$, paired $t$-test $p = 0.033$; false alarm rate, light-off trials: $0.37 \pm 0.02$, light-on trials: $0.59 \pm 0.03$, paired $t$-test $p = 4.5 \times 10^{-10}$; catch rate, light-off trials: $0.29 \pm 0.02$, light-on trials: $0.43 \pm 0.03$, paired $t$-test $p = 2.5 \times 10^{-6}$) (distractor-aligned wMC suppression, $n = 34$ sessions, hit rate: light-off trials: $0.90 \pm 0.02$, light-on trials: $0.93 \pm 0.01$, paired $t$-test $p = 0.017$; false alarm rate, light-off trials: $0.33 \pm 0.03$, light-on trials: $0.52 \pm 0.04$, paired $t$-test $p = 1.3 \times 10^{-7}$; catch rate, light-off trials: $0.29 \pm 0.03$, light-on trials: $0.40 \pm 0.03$, paired $t$-test $p = 0.0021$). Similar effects were observed for small amplitude stimuli (Supplementary Fig. 2).

Increases in response rates can reflect increased stimulus detection and/or increased tendency to respond. Therefore, we used signal detection theory to transform response rates into detection (behavioral d prime, $d'$) and tendency to respond (criterion, c) (Fig. 1g bottom and Supplementary Fig. 3). wMC suppression increased the tendency to respond (reduction in c) (target-aligned wMC, light-off trials: $0.46 \pm 0.06$, light-on trials: $-0.03 \pm 0.07$, paired $t$-test $p = 1.6 \times 10^{-10}$; distractor-aligned wMC, light-off trials: $0.53 \pm 0.09$, light-on trials: $0.07 \pm 0.08$, paired $t$-test $p = 4.4 \times 10^{-7}$, average decrease of 0.47) and increased distractor detection (increase in behavioral $d'$) (target-aligned wMC, light-off trials: $0.21 \pm 0.04$, light-on trials: $0.42 \pm 0.07$, paired $t$-test $p = 0.019$; distractor-aligned wMC, light-off trials: $0.09 \pm 0.06$, light-on trials: $0.37 \pm 0.08$, paired $t$-test $p = 0.0079$, average increase of 0.27) but did not increase target detection (target-aligned wMC, light-off trials: $2.17 \pm 0.13$, light-on trials: $1.71 \pm 0.11$, paired $t$-test $p = 3.6 \times 10^{-4}$, significantly decreased; distractor-aligned wMC, light-off trials: $1.95 \pm 0.14$, light-on trials: $1.63 \pm 0.11$, paired $t$-test $p = 0.20$) From these analyses we conclude that in expert mice wMC suppresses both the tendency to respond and the detection of distractor stimuli.

We conducted multiple control experiments to determine the specificity of this effect. First, we applied the same optogenetic suppression to target-aligned S1 and distractor-aligned S1 (Fig. 1h, Supplementary Figs. 2b, 3). Suppression of target-aligned S1 resulted in trends towards reduced response rates, with significant reductions in hit rate, false alarm rate, and target detection for small amplitude stimuli (Supplementary Fig. 2b). Reduced responding during target-aligned S1 suppression contrasts with increased responding during wMC suppression, suggesting opposing functional contributions to task performance. Second, we applied the same optogenetic protocols to wild type (WT) littermates that lacked channelrhodopsin (ChR2) expression. We did not find robust differences between light-on and light-off trials (Fig. 1i, Supplementary Figs. 2c, 3) indicating that the behavioral effects from wMC and S1 suppression are not due to artifacts of opto-stimulation.

We were surprised to find that suppression of target-aligned wMC and distractor-aligned wMC had such similar behavioral effects. We wondered if this could be explained by their strong inter-hemispheric connections[27]. Indeed, we found that indirect suppression of target-aligned wMC (light on distractor-aligned wMC) was ~50% as effective as direct suppression (light on target-aligned wMC) (reduction in putative excitatory neuron spike rates: direct, $74 \pm 11\%$; indirect, $36 \pm 10\%$, Supplementary Fig. 1). Thus, the behavioral effects observed from unilateral wMC suppression may reflect disruptions of bilateral coordination.

### Asymmetric cross-hemispheric propagation of target and distractor stimuli during task engagement

To determine the neuronal mechanisms of these behavioral effects, we began by characterizing sensory processing within S1. We recorded spiking activity of single units in layer 5 S1 during task performance (Fig. 2a, b). First, we analyzed population (summation of single units recording simultaneously in each session) whisker stimulus encoding using neurometric analyses (Fig. 2c). We refer to preferred stimuli (target stimuli for target-aligned S1, distractor stimuli for distractor-aligned S1) and unpreferred stimuli (distractor stimuli for target-aligned S1, target stimuli for distractor-aligned S1; also referred to as "cross-hemispheric" or "propagated" stimuli). In awake behaving mice during task engagement, target-aligned and distractor-aligned S1 encoded their preferred stimuli robustly and at short latency, peaking at 40-45 ms post-stimulus (Fig. 2d, e). As shown in the neuronal d prime ($d'$) plots (Fig. 2e, middle), these signals overlap until ~40 ms, after which the target stimulus encoding remains elevated and the distractor stimulus encoding decreases back to baseline. Notably, the divergence in stimulus encoding temporally correlated with the onset

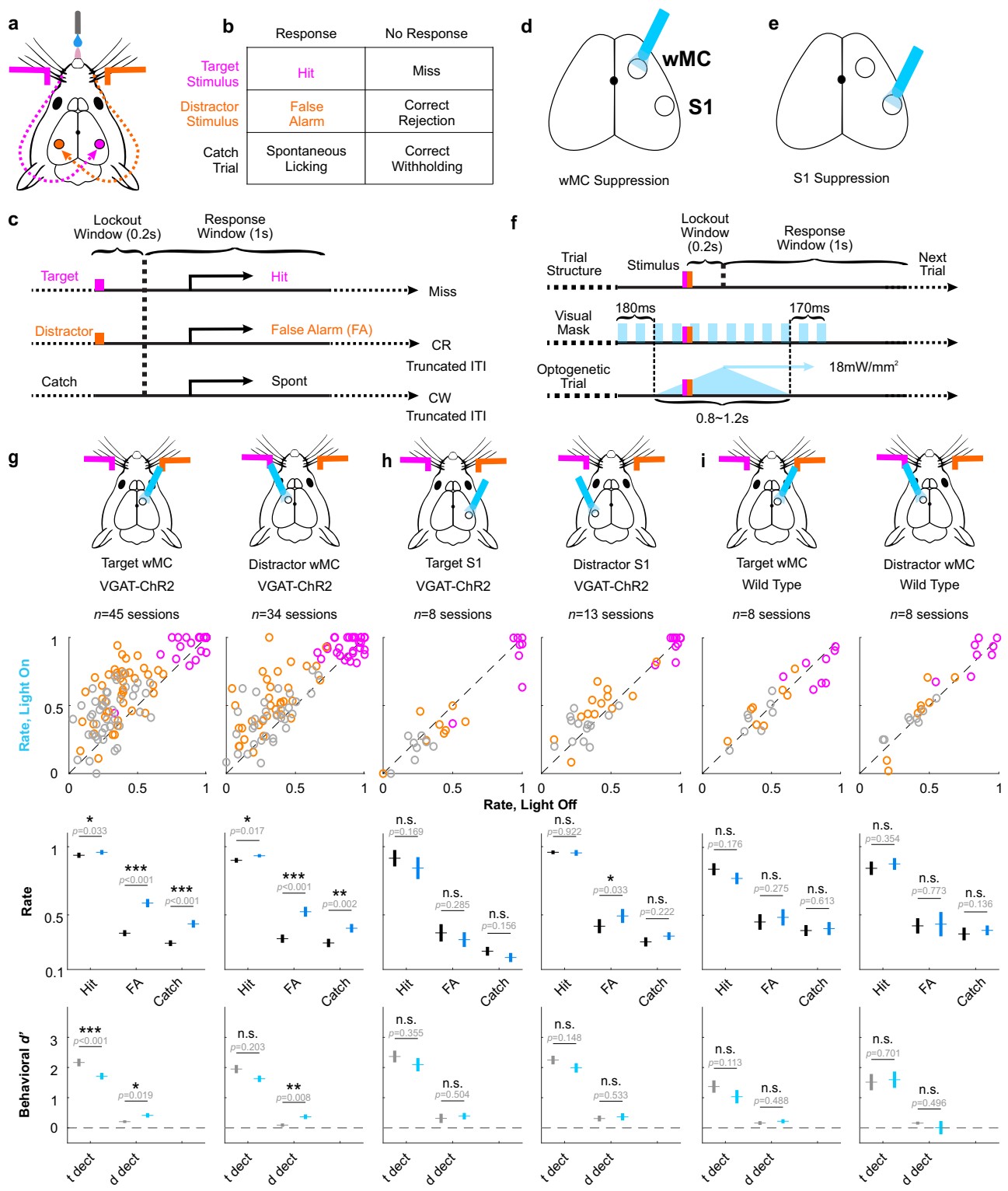

of increased whisker movements on target trials (Fig. 2e, bottom). Thus, the >40 ms persistent activity in target-aligned S1 on target trials may be accounted for, at least in part, by the onset of whisking before the response window (which starts at 200 ms post-stimulus).

We also quantified the propagation of target and distractor stimuli across hemispheres (Fig. 2f, g). Distractor-aligned S1 showed robust encoding of unpreferred (target) stimuli (Fig. 2g), peaking at ~60–80 ms post-stimulus. This longer latency is consistent with a longer pathway for unpreferred (ipsilateral) stimuli to reach S1. In marked contrast, we observed a slight negative encoding of distractor

stimuli in target-aligned S1 (Fig. 2g) (−0.02 ± 0.02, neuronal $d'$ averaged over the 100 ms window, $n = 17$ sessions; one sample $t$-test for all 17 sessions in each 20 ms sliding window, 60–80 ms: $p = 0.045$; 65–85 ms: $p = 0.040$). Accordingly, the propagation of target stimuli into distractor-aligned S1 was significantly larger than distractor stimuli into target-aligned S1 (neurometric averaged across 100 ms, two sample $t$-test, $p = 9.5 \times 10^{-7}$). Moreover, unlike the divergence of encoding of preferred stimuli (Fig. 2e, middle), the divergence of unpreferred stimuli (Fig. 2g, middle) occurred immediately after stimulus onset and was statistically significant within 20 ms post-

**Fig. 1 | Suppressing either target-aligned or distractor-aligned wMC increases behavioral responses. a** Illustration of the behavioral task setup. Mice are head-fixed in the behavioral rig with piezo-controlled paddles within their whisker fields bilaterally. Whisker stimulus-triggered neuronal responses propagate to S1 in the contralateral hemisphere. **b** Task trial types and outcomes: for each trial there can be a target stimulus (hit or miss), a distractor stimulus (false alarm, FA, or correct rejection, CR), or no stimulus catch trial (spontaneous licking or correct with-holding, CW). **c** Task structure: after whisker stimulus onset there is a 0.2 s lockout window, followed by a 1 s response window. Responding outside of the response window is punished by a time-out (resetting the inter-trial interval, ITI). **d**, **e** Illustration of the optogenetic suppression design, in VGAT-ChR2 transgenic mice. Optogenetic suppression (blue light-on) of wMC or S1 was performed on 33% of trials, randomly interleaved with control (light-off) trials. **f** Features of optogenetic suppression and visual mask. **g** Behavioral responses for optogenetic light-on trials and control light-off trials across all sessions for wMC suppression (target-aligned: 1st column; distractor-aligned:2nd column). The top row indicates the site of optogenetic suppression. The first data row displays hit rates (magenta), false alarm rates (orange), and catch rates (gray), each circle reflecting one behavioral session. The middle data row displays the average hit rate, false alarm rate and catch rate. The bottom row displays the average behavioral *d'* for target detection and distractor detection (see Methods). For bottom rows, data from light-off control trials are in black and data from light-on optogenetic suppression trials are in blue. Data are represented as mean ± the standard error of the mean (SEM). Results from two-tailed paired *t*-tests are reported, without correction for multiple comparisons. Exact *p*-values are shown in the figure. Throughout the article, *p* values are indicated as: n.s. ($p \geq 0.05$), *($0.01 \leq p < 0.05$), **($0.001 \leq p < 0.01$), or ***($p < 0.001$). **h** As same as [**g**] but for optogenetic suppression of S1. **i** As same as [**g**] but for optogenetic light above wMC in wild type mice not expressing ChR2.

stimulus. Could this difference in stimulus propagation be accounted for by whisker movements? Distractor-aligned whisker movements did significantly increase after target stimulus onset (Fig. 2g, bottom). However, this increase lagged the spiking increase in distractor-aligned S1. Thus, target stimuli propagate across hemispheres while distractor stimuli do not, and this difference emerges before differences in both preferred stimulus encoding and whisker movements.

We further characterized the effects of trial outcome, behavioral state, and training on stimulus propagation. Lack of distractor propagation to target-aligned S1 was present on both false alarm and correct rejection trials (Fig. 2h), although increases in target-aligned S1 spiking was observed on false alarm trials during the response window (Supplementary Fig. 4). Next, we determined which aspects of preferred and unpreferred stimulus encoding were dependent on task engagement. Neuronal activities during task disengagement were obtained from the same expert mice, after they stopped responding within a session (presumably due to satiety). Generally, disengagement led to reduced stimulus encoding (Fig. 2i–l). This reduction was statistically significant for target stimuli in target-aligned S1 (Fig. 2i) and trending for distractor stimuli in distractor-aligned S1 (Fig. 2k) and target stimulus propagation into distractor-aligned S1 (Fig. 2l). In marked contrast, disengagement led to a significant increase in distractor stimulus propagation into target-aligned S1 (Fig. 2j). Additionally, we recorded S1 responses to preferred and unpreferred whisker stimuli in untrained (naive) anesthetized mice and in awake mice that were habituated to head-fixation but naive to the whisker detection task (Supplementary Fig. 5c, d). We compared the sensory-evoked cross-hemispheric (unpreferred) propagation from four conditions: expert target, expert distractor, naive awake, and naive anesthetized (Supplementary Fig. 5e, f). We found that in both naive awake and naive anesthetized mice there is modest cross-hemispheric propagation. This contrasts with expert mice which show robust propagation of target stimuli and suppressed propagation of distractor stimuli. Thus, behavioral training results in a bidirectional change in cross-hemispheric propagation. In summary, we observed a strong asymmetry in the propagation of target and distractor stimuli across hemispheres, with a suppression of distractor stimulus response propagation into target-aligned S1 (Fig. 2g) that is dependent on task learning and task engagement.

To verify this observation using a different recording method, we analyzed widefield Ca²⁺ -sensor imaging data from different mice performing the same behavioral task[7]. We observed a similar asymmetric cross-hemispheric propagation in our imaging data (Supplementary Fig. 6). We found that target whisker stimuli induced significant activation of distractor-aligned S1 ($n = 40$ sessions, neuronal *d'*: $0.14 \pm 0.03$, one sample *t*-test: $p = 0.0002$). In contrast, distractor stimuli induced significant suppression of target-aligned S1 ($n = 40$ sessions, neuronal *d'*: $-0.07 \pm 0.02$, one sample *t*-test: $p = 0.007$; paired *t*-test comparing target and distractor propagation, $p = 1.8 \times 10^{-5}$).

## Changes in S1 sensory-evoked responses during wMC suppression

Does wMC contribute to the task-related sensory processing adaptations described above? One of the main output projections from wMC is to S1, through both the direct cortical feedback pathway and indirect cortico-thalamo-cortical pathways[26,27,35–37] (see also Supplementary Fig. 7). We sought to determine the consequences of wMC suppression on sensory encoding in S1. We recorded S1 spiking activity as described above while also applying interleaved optogenetic wMC suppression. Target-aligned S1 recordings were paired with target-aligned wMC suppression (Fig. 3a) and distractor-aligned S1 recordings were paired with distractor-aligned wMC suppression (Fig. 3d). wMC suppression reduced S1 baseline firing rates (characterized further below and in Fig. 5). Here we focus on the effects of wMC suppression on S1 stimulus encoding, as determined by population neurometric analyses.

First, we report the effects of wMC suppression on S1 stimulus encoding by comparing optogenetic light-off and light-on conditions. wMC suppression transiently increased target stimulus encoding in target-aligned S1 (Fig. 3b, middle), which we interpret as wMC marginally suppressing target stimulus encoding. wMC suppression did not significantly impact distractor-aligned S1 encoding of distractor stimuli (Fig. 3e, middle) or the propagation of target stimuli into distractor-aligned S1 (Fig. 3f, middle). However, wMC suppression did significantly increase the propagation of distractor stimuli into target-aligned S1 (Fig. 3c, middle) (two sample *t*-test, $p < 0.05$ during 30–100 ms post stimulus window). This effect was present from 30–100 ms post-stimulus, well before the response time (minimum 200 ms), and similar to the latency of unpreferred stimulus encoding in distractor-aligned S1. Importantly, this increase in target-aligned S1 encoding of distractor stimuli was present regardless of trial outcome (false alarm or correction rejection, Fig. 3g–i) and was not present with optogenetic suppression alone (i.e., on catch trials) (Fig. 3j–l).

As described above, task-learning and engagement increases cross-hemispheric propagation of target stimuli and decreases cross-hemispheric propagation of distractor stimuli (Supplementary Fig. 5). It appears that in expert mice wMC actively contributes to one aspect of this bidirectional modulation: the suppression of distractor propagation. Moreover, these neurometric findings are consistent with, and indeed may contribute to, increased distractor detection during wMC suppression (Fig. 1g). We further explored this potential relationship through neurometric-psychometric comparisons[38]. Target-aligned wMC suppression caused an average increase in behavioral distractor detection (behavioral *d'*) of $0.22 \pm 0.08$ which precisely matches the average increase in distractor encoding (neuronal *d'*) in target-aligned S1 of $0.24 \pm 0.05$. Second, in a session-by-session analysis, increased distractor encoding in target-aligned S1 had a positive correlation trend with increased behavioral distractor detection ($R^2 = 0.12$) in contrast to a near zero correlation with

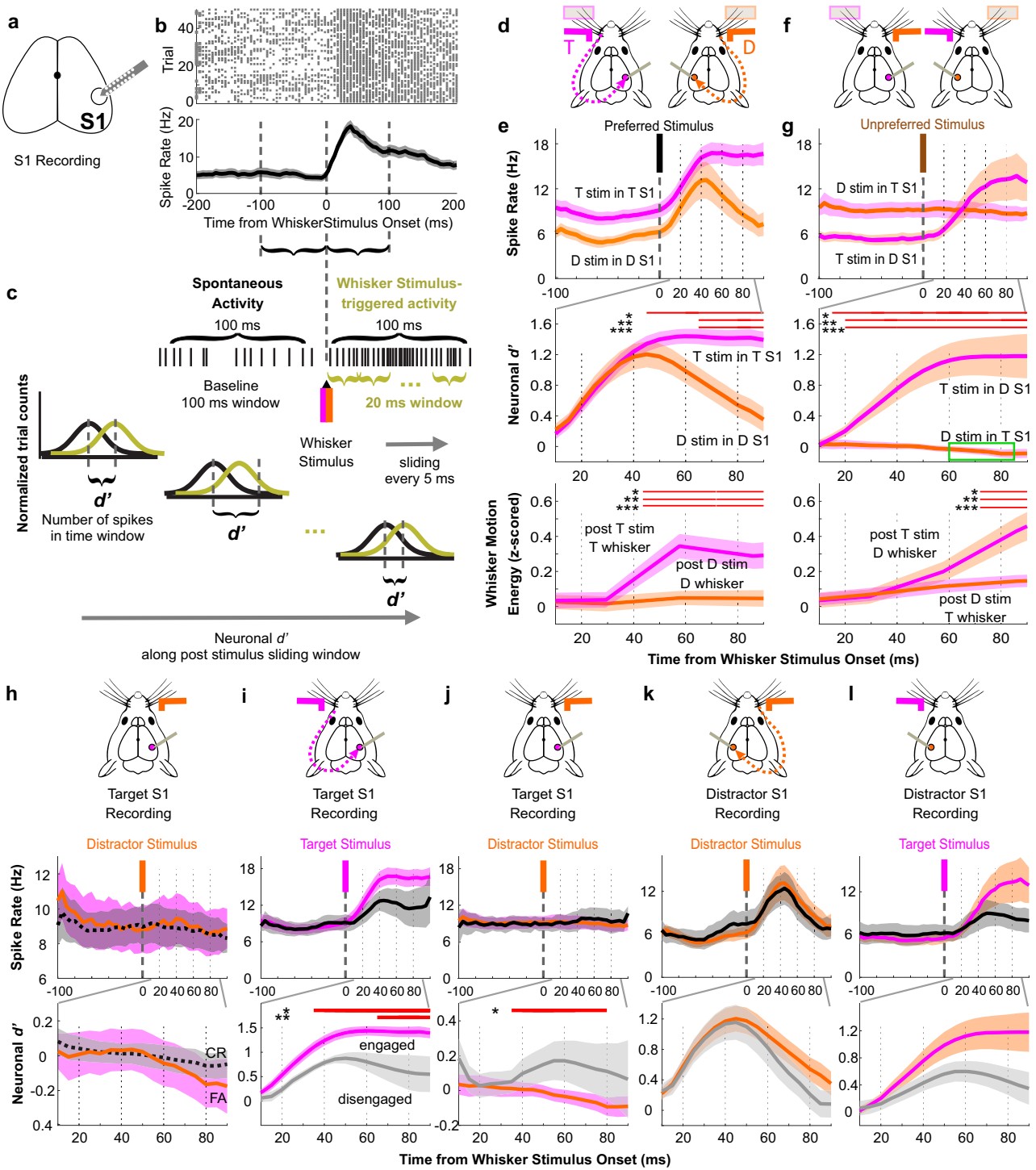

changes in the tendency to respond ($R^2 = 0.0035$) (Supplementary Fig. 8). These analyses suggest that wMC suppression of distractor propagation into target-aligned S1 reduces distractor-evoked behavioral responses.

Simultaneous whisker imaging data suggest that these changes in stimulus encoding do not simply reflect changes in whisker motion. wMC suppression did not consistently impact whisker motion energy, either before whisker stimulus onset (Supplementary Fig. 9) or during the early post-stimulus window (0–100 ms) (Fig. 3b, c, e, f, bottom). Importantly, increased distractor stimulus propagation into target-aligned S1 during wMC suppression cannot be simply accounted for by changes in whisker motion energy (Fig. 3c).

## wMC decorrelates target and distractor stimulus encoding in target-aligned S1 excitatory neurons

In the analyses described above we report population activity (from summed single units). Here, we report single unit analyses, segregated into putative excitatory and inhibitory neurons based on opto-tagging (Fig. 4a and Supplementary Fig. 10). Below, we focus our analyses on putative excitatory neurons, as the output of S1.

First, we present the activity of one example excitatory neuron in target-aligned S1 (Fig. 4b). In control (light-off) conditions (Fig. 4b, left), this neuron responds robustly to target stimuli, but not to distractor stimuli. During wMC suppression (Fig. 4b, right), responses to target stimuli are maintained, yet now we observe increased responses

**Fig. 2 | Asymmetric cross-hemispheric propagation of target and distractor whisker stimulus responses during task performance. a** Illustration of extracellular silicon probe recording of S1. **b** Example session of S1 multi-unit spiking activity in response to preferred (contralateral) whisker stimuli, from one awake behaving recording session. Top plot: raster plots of neuronal spiking for multiple trials. Bottom plot: averaged from the data above, represented as mean ± SEM. **c** Illustration of neuronal $d'$ calculation (see Methods). **d–h** Analyses of recordings in expert mice during task performance. For colored traces, the solid line reflects the stimulus identity (target, magenta; distractor, orange) whereas the SEM shading reflects the recording side of S1 or whisker field (target, magenta; distractor, orange). **d** Experimental design illustration for [**e**]: recordings of S1 and whisker fields aligned to their preferred stimuli. **e** Neuronal spike rate (top), neuronal $d'$ (middle) and whisker motion energy (bottom). For abbreviations: "T stim" or "D stim": target or distractor whisker stimulus; "T S1" or "D S1": target-aligned or distractor-aligned S1; "T whisker" or "D whisker": target or distractor whisker field.

Data are grand averages, combined from all recording sessions. The red bars indicate the epochs in which neuronal $d'$ (middle) or whisker motion energy (bottom) from the two recording-stimulation conditions are significantly different from each other (statistical comparisons of spike rate data were not conducted here); same indication for [**g**, **i–l**]. **f** Experimental design illustration for [**g**]: recordings of S1 and whisker fields contralateral to their preferred stimuli (unpreferred stimuli). **g** Same general layout as [**e**]. The green box indicates the epochs in which neuronal $d'$ for distractor stimulus encoding in target-aligned S1 is significantly below zero. **h** Comparison of distractor whisker stimulus responses in target-aligned S1 based on trial outcome (FA: color traces; CR: black traces). **i–l** Comparisons of S1 neuronal activity between engaged (colored) and disengaged (black/gray) recording sessions. **i** Neuronal spike rate (top) and neuronal $d'$ (bottom) of preferred whisker stimuli in target-aligned S1 (**j**) As same as [**i**] but for unpreferred stimuli in target-aligned S1. **k**, **l** As same as [**i**, **j**] but for distractor-aligned S1 recordings.

to distractor stimuli (change in spike rate following distractor whisker stimuli: control, 0.36 Hz decrease; wMC suppression, 4.2 Hz increase; $p = 0.14$). We interpret this finding as reduced stimulus selectivity, in which this neuron increases its apparent receptive field size to now include distractor (unpreferred) inputs. From this perspective, an important function of wMC is to gate out distractor stimulus responses from target-aligned S1 neurons.

We recognize two potential mechanisms by which wMC may gate distractor stimulus responses in target-aligned S1 (Fig. 4c). First, there could be a shift of the entire population, such that throughout the distribution of S1 neurons there is a proportional reduction of distractor encoding (gating model 1, Fig. 4c, middle). Second, there could be a rotation in the distribution of S1 neurons, such that encoding of target and distractor stimuli are decorrelated in individual neurons (gating model 2, Fig. 4c, bottom). To distinguish between these mechanisms, we plotted the distribution of preferred (contralateral) and unpreferred (ipsilateral) stimulus encoding from target-aligned S1 neurons, distractor-aligned S1 neurons, and from S1 neurons in naive, anesthetized mice (Fig. 4d–g). We first describe stimulus selectivity profiles in control light-off conditions. In anesthetized mice (Fig. 4d, g, green), S1 neurons displayed a slight positive correlation, such that the neurons with the strongest encoding of preferred (contralateral) stimuli also positively encoded unpreferred (ipsilateral) stimuli (95% confidence interval of the slope of the linear fit: 0.07–0.21, for $n = 200$ single units). During task performance, distractor-aligned and target-aligned S1 neurons displayed remarkably different stimulus selectivity profiles. Distractor-aligned S1 neurons were less selective than the anesthetized population, with a more positive slope to the linear fit of their stimulus response correlations (95% confidence interval of the slope of the linear fit: 0.23–0.57, for $n = 117$ single units) (Fig. 4e, g, orange). In contrast, target-aligned S1 neurons were more selective than the anesthetized population, with a flat, trending to negative, slope to the linear fit of their stimulus response correlations (95% confidence interval of the slope of the linear fit: −0.10–0.01, for $n = 197$ single units) (Fig. 4f, g, magenta). That is, across target-aligned S1 neurons, target and distractor stimulus encoding is decorrelated.

Next, we tested the impact of wMC suppression on stimulus selectivity, starting with target-aligned S1. As shown in Fig. 4f (blue), wMC suppression caused a rotation in the stimulus selectivity of target-aligned S1 neurons, such that now there was a positive slope to the population stimulus response correlation (95% confidence interval of the slope of the linear fit: 0.08–0.19). Thus, wMC contributes to distractor response suppression in target-aligned S1 by modulating single unit sensory selectivity. This rotation in stimulus correlation space (Fig. 4f bottom) is consistent with the "gating 2" model shown in Fig. 4c, as opposed to a shift in distractor stimulus encoding across the population ("gating 1" model).

wMC suppression also caused a positive rotation in the stimulus selectivity space for the other conditions: significant for anesthetized

recordings (Fig. 4d), and a non-significant trend for distractor-aligned S1 (Fig. 4e). These data may suggest that suppression of unpreferred stimuli is a general, non-context dependent, function of wMC. However, further analyses indicate that stimulus selectivity and its modulation by wMC is indeed task modulated and context dependent. First, the effects of wMC suppression were different for putative inhibitory neurons (Supplementary Fig. 11a, b). In anesthetized mice, there was a trend for wMC suppression to increase the slope of stimulus correlation (reflecting reduced selectivity) of putative inhibitory neurons. Yet, there was a trend in the opposite direction for putative inhibitory neurons in target-aligned and distractor-aligned S1 during task performance (reduced slope, increased stimulus selectivity) (Supplementary Fig. 11a). Moreover, when mice were disengaged from the task, differences in stimulus selectivity were less pronounced between target-aligned and distractor-aligned S1 (Supplementary Fig. 11c-f). Importantly, when mice were disengaged, wMC suppression in target-aligned S1 did not change stimulus selectivity (Supplementary Fig. 11c). Altogether, these data suggest that asymmetric modulation of stimulus selectivity in S1 is a robust feature of selective detection, and that wMC contributes to this modulation.

The wMC-mediated decorrelation of target and distractor encoding in target-aligned S1 excitatory neurons is likely to be critical for distractor response suppression in this task. During wMC suppression, due to the positive correlation across single units for sensory encoding of target and distractor stimuli, it would be difficult for a down-stream reader to distinguish between a small target stimulus and a large distractor stimulus, and thus could contribute to increased false alarm rates. We tested this hypothesis using a linear classier to mimic the downstream reader. First, we chose the half of the target-aligned S1 excitatory neurons with the largest target stimulus encoding, as the putative population signaling target detection to the downstream region. Next, we trained a classifier on single trial data from this ensemble, to distinguish between target and distractor trials. We then tested the classifier on both hold-out control trials and wMC suppression trials, from the same neuronal population (see Methods). We found that this simulated downstream reader was nearly perfect at correctly classifying control data as target and distractor trials (error rate, 0.55%). In marked contrast, the downstream reader error rate was substantial for wMC suppression data (20.31%, Supplementary Fig. 12). Notably, the error types for wMC suppression data partially matched the behavioral data, demonstrating higher "false alarm rates" (mis-classification of distractor as target trials) than "miss rates" (mis-classification of target as distractor trials) (30.09% vs 10.52%, Supplementary Fig. 12).

## wMC proactively modulates S1 activity in a context-dependent manner

Top-down modulations of sensory cortex can occur in two fundamentally different ways. Top-down modulations may be stimulus-evoked and reactively exert a functional feedback signal[19,26,39–41],

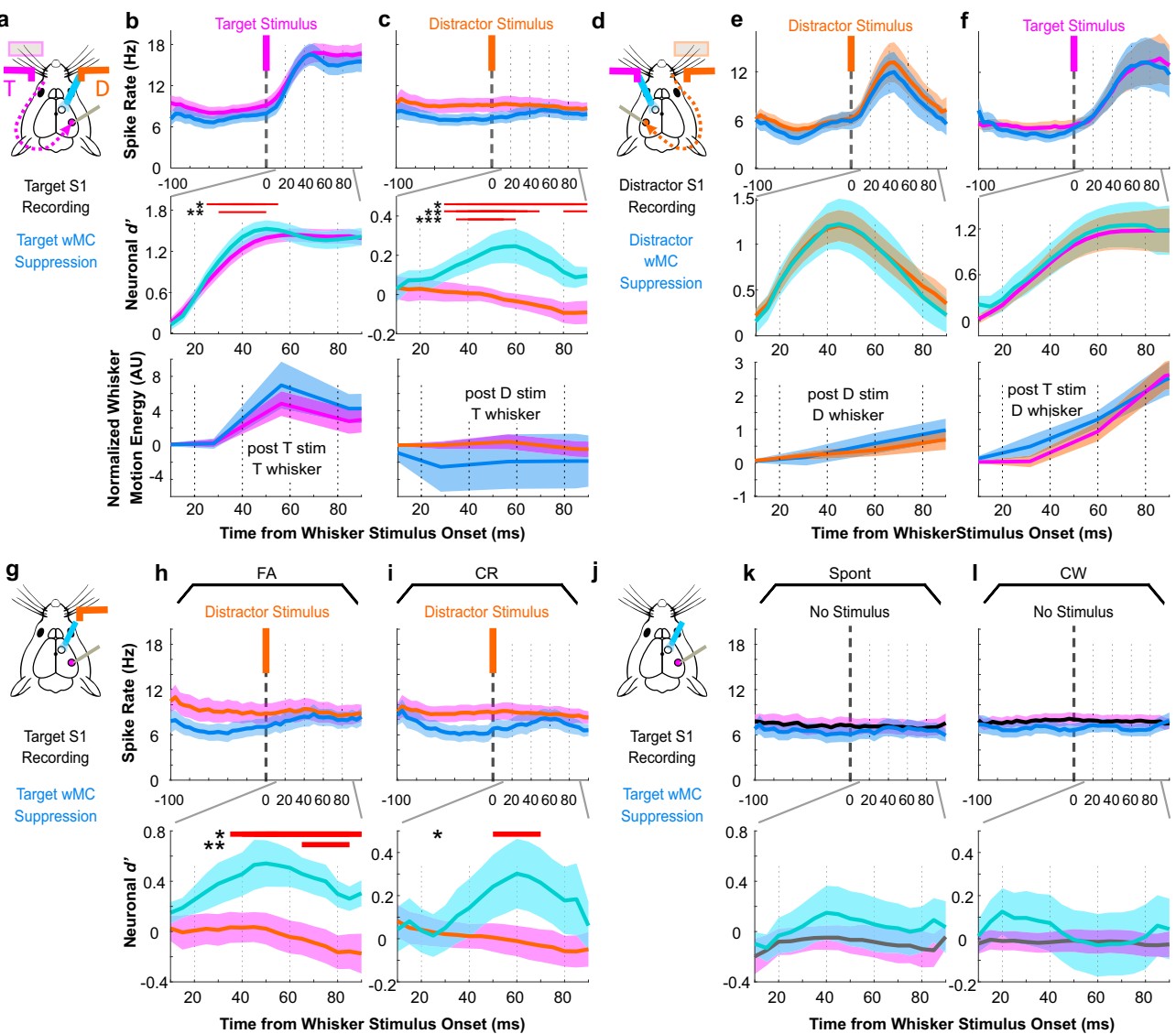

**Fig. 3 | Suppressing wMC increases distractor stimulus encoding in target-aligned S1.** Analyses of neuronal recordings in expert mice during task performance. For magenta/orange traces, the solid line reflects the stimulus identity (target, magenta; distractor, orange) whereas the SEM shading reflects the recording site (target-aligned S1 or target field whiskers, magenta; distractor-aligned S1 or distractor field whisker, orange), as in Fig. 2. **a** Experimental design illustration for [**b**, **c**]: recording in target-aligned S1 or target whisker fields, suppression of target-aligned wMC, target stimulus is preferred and distractor stimulus is unpreferred. **b**, **c** Neuronal spike rate (top) and neuronal $d'$ (middle) of target-aligned S1 neurons, and whisker motion energy within the target whisker field (bottom) during control trials (light-off, magenta) and wMC suppression trials (light-on, blue) in response to target [**b**] or distractor [**c**] whisker stimuli. Data are grand averages, combined from all recording sessions. The red bars indicate the epochs in which neuronal $d'$

(middle) or whisker motion energy (bottom) during wMC suppression is significantly different from control trials (statistical comparisons of spike rate data were not conducted here). 'AU', arbitrary unit. **d** Experimental design illustration for [e,f]: recording in distractor-aligned S1 or distractor whisker fields, suppression of distractor-aligned wMC, distractor stimulus is preferred and target stimulus is unpreferred. **e**, **f** Same general layout as [**b**, **c**] but for recordings of distractor-aligned S1 and distractor whisker fields during distractor-aligned wMC suppression. **g**–**l** Data for spike rates and neuronal $d'$ are presented as above. **g** Diagram of target-aligned S1 recordings and target-aligned wMC suppression for distractor trials. Comparison of wMC suppression and control conditions on false alarm (FA) trials (**h**) and correct rejection (CR) trials (**i**). **j**–**l** Same as [**g**–**i**] but for catch trials without whisker stimuli, showing spontaneous responding (Spont) and correct withhold (CW) trials. Data represented as mean ± SEM.

Alternatively, top-down modulations may be in place before a stimulus arrives, proactively setting the initial condition for goal-directed sensory processing[42]. These last sets of experiments and analyses were to determine whether there is evidence for proactive (pre-stimulus) wMC modulation. We begin with analyses of S1 single unit spike rates (SR). For these analyses, we considered the last 100 ms pre-stimulus (Fig. 5a), in which we could achieve a relatively stable baseline during wMC optogenetic suppression. We calculated the pre-stimulus modulation index for each neuron as $MI = (light\_on\_SR - light\_off\_SR) / (light\_on\_SR + light\_off\_SR)$. As such, $MI < 0$ indicates that wMC

suppression reduces spiking (alternatively, that wMC normally drives spiking) in that neuron.

For recording conditions of anesthetized, awake behaving target-aligned, and awake behaving distractor-aligned, wMC robustly drives pre-stimulus spiking in S1 neurons (Fig. 5b–d). This effect was not present in wild type control mice lacking channelrhodopsin expression, and therefore is unlikely to be due to artifacts of optogenetic stimulation (modulation index (MI) of each group, one sample t-test p-values of each group, and two sample t-test p values between groups: wild type: 0.02 ± 0.02, p = 0.21; VGAT-ChR2:

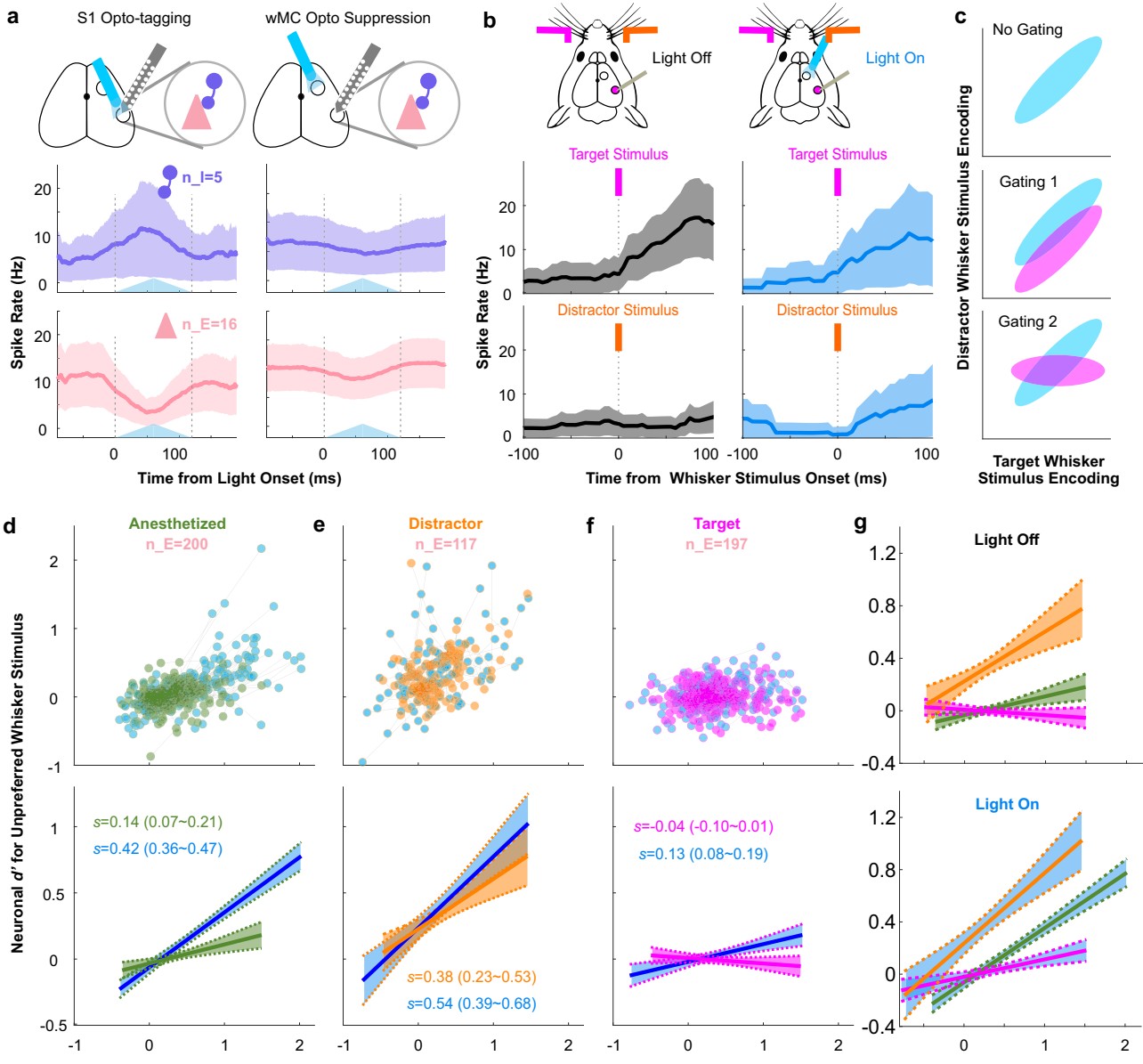

**Fig. 4 | Asymmetric stimulus selectivity of S1 single units and their modulation by wMC. a** Example responses of S1 putative excitatory (pink) and inhibitory (purple) neurons to S1 opto-tagging (left) and the same neurons to wMC optogenetic suppression (right). Top row: for S1 opto-tagging, a light fiber is positioned above the S1 recording probe. For wMC suppression, a second light fiber is positioned above wMC. Middle and bottom plots: average spiking activity recorded from one example awake behaving session with recording/suppression of the distractor-aligned hemisphere. The blue triangles reflect the time course of opto-stimulation. **b** Target and distractor whisker stimulus evoked responses of an example putative excitatory neuron in target-aligned S1. Left: light-off control trials; right: light-on wMC suppression trials. For [a,b], data are represented as mean ± SEM. **c** Illustration of potential mechanisms of distractor whisker stimulus gating plotted in stimulus selectivity space, with target whisker stimulus encoding on the x-axis and distractor whisker stimulus encoding on the y-axis. Blue distributions reflect a population of target-aligned S1 neurons without distractor gating. Magenta distributions reflect the same population of neurons with distractor gating. **d**–**f** Putative excitatory neurons recorded (**d**) under anesthesia, (**e**) in distractor-aligned S1 in awake behaving mice, and (**f**) in target-aligned S1 in awake behaving mice. For [**d**–**f**], the top scatter plot displays single neuron $d'$ values for their preferred whisker stimulus (contralateral to the recording site, x-axis) vs their unpreferred whisker stimulus (ipsilateral to the recording site, y-axis). The green, orange, and magenta dots indicate the neuronal $d'$ values with wMC intact (control, light-off) and the blue dots indicate the neuronal $d'$ values of the same neurons during wMC suppression (light-on). Gray lines connect the same neurons under light-off and light-on conditions. The bottom plots show the linear fits of the single unit data above (solid lines: linear fit; dashed lines: 95% confidence intervals; $s$ = slope of linear fit (95% confidence intervals)). **g** The overlay of the light-off control (top) and light-on wMC suppression (bottom) linear fits, re-plotted from the bottom row of [**d**–**f**].

−0.21 ± 0.01, $p = 7.8 \times 10^{-112}$; wild type vs VGAT-ChR2: $p = 6.0 \times 10^{-14}$) (Fig. 5b). Interestingly, wMC modulation of S1 pre-stimulus activity under anesthesia was larger than during task performance (*MI* of each group, one sample *t*-test *p*-values of each group, and two sample *t*-test *p*-values between groups: anesthesia: −0.42 ± 0.02, $p = 4.9 \times 10^{-82}$; awake behaving: −0.12 ± 0.01, $p = 1.4 \times 10^{-26}$; anesthesia

vs awake behaving: $p = 4.7 \times 10^{-49}$) (Fig. 5c). The modulation indices were similar in awake behaving target-aligned and distractor-aligned S1 (*MI* of each group, one sample *t*-test *p*-values of each group, and two sample *t*-test *p*-values between groups: target: −0.12 ± 0.01, $p = 2.0 \times 10^{-20}$; distractor: −0.13 ± 0.02, $p = 8.5 \times 10^{-9}$; target vs distractor: 0.65) (Fig. 5d), indicating similar levels of overall wMC drive.

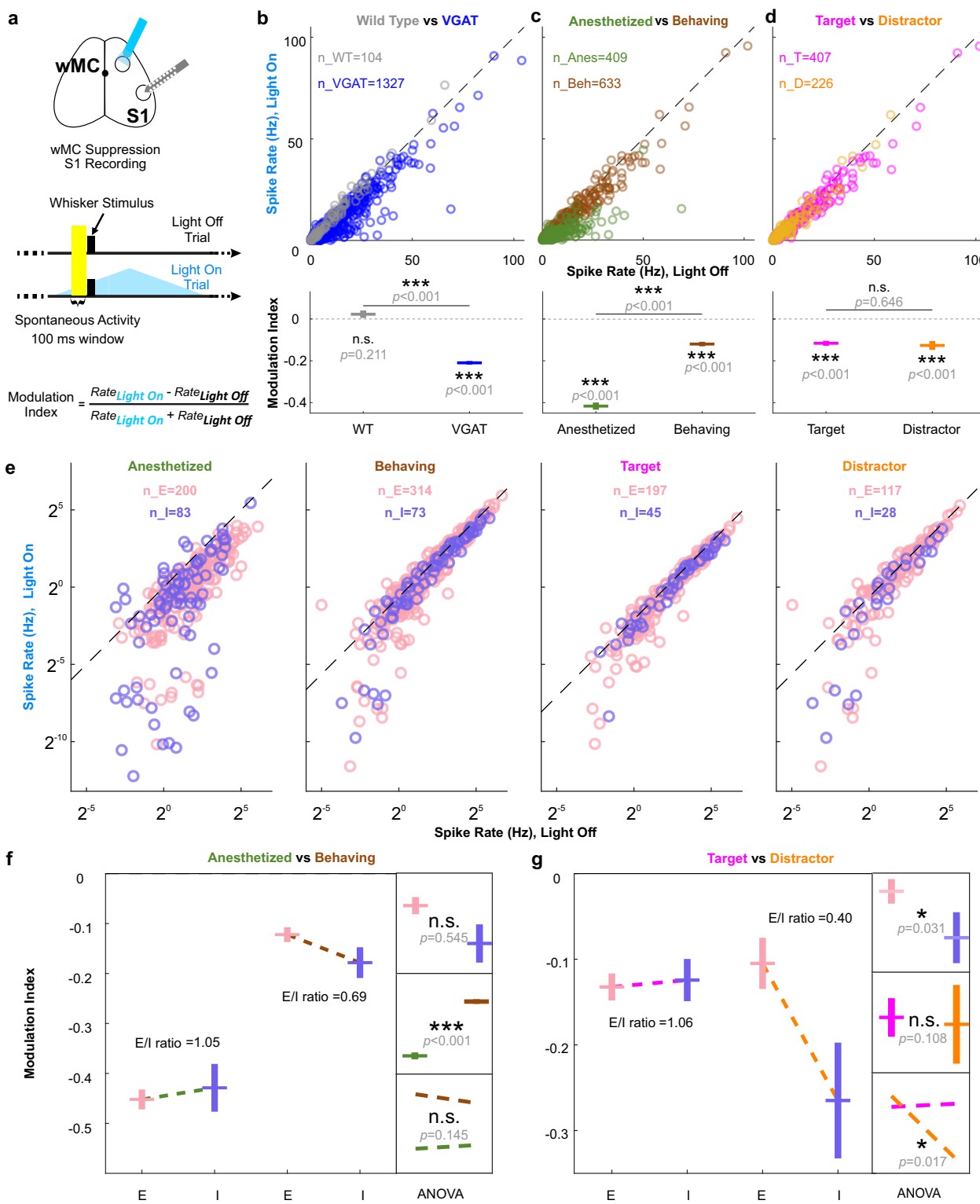

To gain further insights into changes in local circuits, we analyzed wMC modulation separately for putative excitatory and inhibitory S1 neurons. Both excitatory and inhibitory S1 neurons were significantly driven by wMC in anesthetized and awake behaving mice, both target-aligned and distractor-aligned (paired $t$-test: anesthetized excitatory: $p = 8.6 \times 10^{-17}$; anesthetized inhibitory: $p = 8.9 \times 10^{-8}$; awake excitatory: $p = 3.1 \times 10^{-12}$; awake inhibitory: $p = 9.7 \times 10^{-6}$; target-aligned excitatory: $p = 1.2 \times 10^{-11}$; target-aligned inhibitory: $p = 2.3 \times 10^{-4}$; distractor-aligned excitatory: $p = 0.022$; distractor-aligned inhibitory: $p = 0.0071$) (Fig. 5e–g). For each condition, we calculated the E/I modulation ratio,

as the *MI* of excitatory neurons divided by the *MI* of inhibitory neurons. E/I ratio for anesthetized and target-aligned conditions were remarkably balanced, yielding ratios near 1. In contrast, the distractor-aligned E/I ratio was 0.40, reflecting a preferential drive of wMC onto putative inhibitory neurons (Fig. 5g). Two-way ANOVA analyses supported these conclusions, yielding a significant main effect for behavioral context comparing anesthetized and awake behaving conditions (anesthetized vs awake, $p = 1.2 \times 10^{-24}$) (Fig. 5f). Comparing target-aligned and distractor-aligned conditions, we found a main effect for cell-type ($p = 0.031$) and an interaction effect between alignment and

**Fig. 5 | wMC proactively drives S1 excitatory and inhibitory neurons in a context-dependent manner. a** Experimental design (S1 opto-tagging was also performed but is not illustrated here). Neuronal activity within the 100 ms pre-stimulus window (highlighted) was used to determine wMC modulation before whisker stimulus onset, assessed in both target-aligned and distractor-aligned hemispheres. **b** Top: S1 pre-stimulus spike rates from wild type (gray) and VGAT-ChR2 (blue) mice. Each circle indicates one neuron. Bottom: modulation index calculated based on the top plot. Above each data point are results from two-tailed one-sample *t*-tests to determine differences from zero within each population, and lines above the data points are results from two-tailed two-sample *t*-tests to determine differences between populations. Corrections for multiple comparisons were not applied. **c** As same as [**b**] but for anesthetized (green) and awake behaving sessions (brown, including both target-aligned and distractor-aligned S1) of VGAT-

ChR2 mice. **d** As same as [**b**] but for awake behaving sessions of VGAT-ChR2 mice in target-aligned S1 (magenta) and distractor-aligned S1 (orange). **e** As same as [**b**−**d**] but separated for excitatory (pink) and inhibitory (purple) neurons under different contexts. **f** Statistical analyses for anesthetized vs awake behaving data. Big box on the left: modulation indices based on the spike rates in [**e**]. Small boxes on the right: results of two-way ANOVA for the modulation indices of anesthesia and awake behaving contexts and excitatory and inhibitory neuron types: top: no significant main effect of neuron type; middle: a significant main effect for context; bottom: no significant interaction effect. **g** Statistical analyses for target-aligned S1 vs distractor-aligned S1 awake behaving data, organized as in [**f**]. ANOVA results: top: a significant main effect for neuron type; middle: no significant main effect of context (alignment); bottom: a significant interaction effect. *p* values are shown in the figure.

cell-type (*p* = 0.017) (Fig. 5g). Consistent with this strong distractor-aligned inhibitory drive, distractor-aligned S1 putative excitatory neurons had a lower pre-stimulus spike rate than target-aligned S1 putative excitatory neurons (target-aligned: 11.33 ± 1.05 Hz, distractor-aligned: 7.56 ± 0.9 Hz, average difference of 3.77 Hz, unpaired *t*-test *p* = 0.016). Notably, this difference was reduced by 29.44% during wMC suppression (target-aligned: 9.74 ± 0.92 Hz, distractor-aligned: 7.08 ± 0.93 Hz, average difference of 2.66 Hz) suggesting that differential E/I drive of wMC onto S1 contributes to differences in pre-stimulus activity. Overall, these data indicate that wMC modulation of S1 is proactive (influencing pre-stimulus activity levels) and context-dependent (for anesthesia vs awake behaving and target vs distractor alignment).

Does proactive modulation contribute to the behavioral effects of wMC suppression? We reasoned that if the behavioral effects are due to proactive modulation, then the period of greatest wMC influence should be leading up to and including the time of whisker stimulus onset. In contrast, a primary reactive function would predict a period of greatest influence to be after stimulus onset. To test this, we conducted optogenetic suppression of wMC in more transient (200 ms) temporal windows (Fig. 6). Three overlapping windows, randomly interleaved in a behavioral session, were (early) exclusively pre-stimulus, (middle) late pre-stimulus and early post-stimulus, or (late) exclusively post-stimulus (Fig. 6d−f, respectively). While all three peri-stimulus suppression windows significantly increased false alarm rates, the only statistically significant increase in distractor detection was from the middle window peaking at the time of whisker stimulus onset (early: *p* = 0.39; middle: *p* = 0.0017; late: *p* = 0.23; Fig. 6e, bottom). As a control for opto-stimulation artifacts, we demonstrate a lack of these effects (increases in false alarm rates and/or distractor detection) in wild type littermates lacking ChR2 expression (Supplementary Fig. 13). We interpret these findings as additional evidence of proactive wMC modulation, setting the initial conditions for the goal-directed routing of feedforward sensory signals.

## Discussion

The major significance of this study is identifying a mechanism by which motor cortex contributes to sensory selection. Within motor cortex, we focused on a region that is part of the sensorimotor cortical whisker system, termed "whisker motor cortex" or wMC. Using opto-genetic suppression, we demonstrated that wMC robustly contributes to behavioral outcomes of suppressing distractor stimulus detection and suppressing the overall tendency to respond (Fig. 1g, Supplementary Fig. 2a, Supplementary Fig. 3). From S1 single unit recordings in behaving mice we revealed an asymmetry in stimulus selectivity, in which distractor-aligned S1 neurons respond to both distractor and target stimuli while target-aligned S1 neurons respond selectively to target stimuli (Figs. 2d−g, 4g). By combining wMC optogenetic suppression with S1 single unit recordings in behaving mice we demonstrated that wMC contributes to target-aligned S1 stimulus selectivity, such that with wMC suppression target-aligned S1 neurons more

strongly encode distractor stimuli (Figs. 3c, 4f). This increase in distractor stimulus encoding matches, and likely underlies, the increase in distractor stimulus detection during wMC suppression (Supplementary Fig. 8). Lastly, we provide evidence that wMC modulation of S1 is context-dependent and proactive (present prior to stimulus onset) (Figs. 5, 6). Our study establishes a tractable and behaviorally-relevant framework for how top-down modulations of sensory propagation implement sensory gating to mediate distractor response suppression.

For combined behavioral-physiological studies, an important consideration is whether the task-related neuronal signals could be fully or partially accounted for by movement[43]. A specific concern for this study is whether the apparent asymmetric cross-hemispheric propagation, and its modulations by wMC, is instead reflecting movement on "Go" (hit, false alarm) trials in our Go/No-Go task design. We believe this is unlikely for four reasons. First, differences in selectivity (between target-aligned S1 and distractor-aligned S1, between control trials and wMC suppression trials) are observable by 20−30 ms post-stimulus onset. This is well before the response time (>200 ms), before the onset of uninstructed whisker movements that are part of the "Go" motor sequence (Figs. 2g, 3c), and well before the emergence of significant choice probability in S1, which includes the contributions of "Go" movements (Supplementary Fig. 4)[44]. Second, increases in distractor responses in target-aligned S1 during wMC suppression are present irrespective of trial outcome (false alarm or correct rejection, Fig. 3g−i), and are not present on catch trials (i.e., in the absence of distractor stimuli) (Fig. 3j−l). Third, target-aligned S1 neurons during task performance are more sensory selective than S1 neurons under anesthesia during which movements are largely suppressed, as well as during task disengagement which lacks responding. Fourth, wMC suppression causes a rotation in stimulus selectivity (Fig. 4f) rather than a shift of the population, as would be expected from increases in arousal or movement[45]. For these reasons we believe that our reported modulations of stimulus selectivity are truly reflecting sensory processing rather than movement confounds. However, we recognize we cannot entirely rule out contributions from body movements or whisker movements below our imaging resolution.

A common framework of top-down cortical modulation is enhancing sensory responses of neurons aligned to the top-down inputs (center) and suppressing sensory responses of neurons unaligned (surround), as measured for the optimal (preferred) stimulus for each population[19,20,39,40,46]. Data from our study, however, do not support this framework. Our data are more consistent with top-down cortical modulation suppressing sensory processing[4,41,47,48], with the suppression of distractor stimulus propagation contributing to preventing behavioral selection of distractor stimuli. This suppressive role of wMC on sensory processing is consistent with the framework of corollary discharge for reafferent stimuli[11,14]. Yet, it suggests a more general role of motor cortices in suppressing sensory processing based on both movement and goal-direction. Additional studies are needed to test this hypothesis across different modalities and behavioral contexts, especially in freely moving behaviors that do

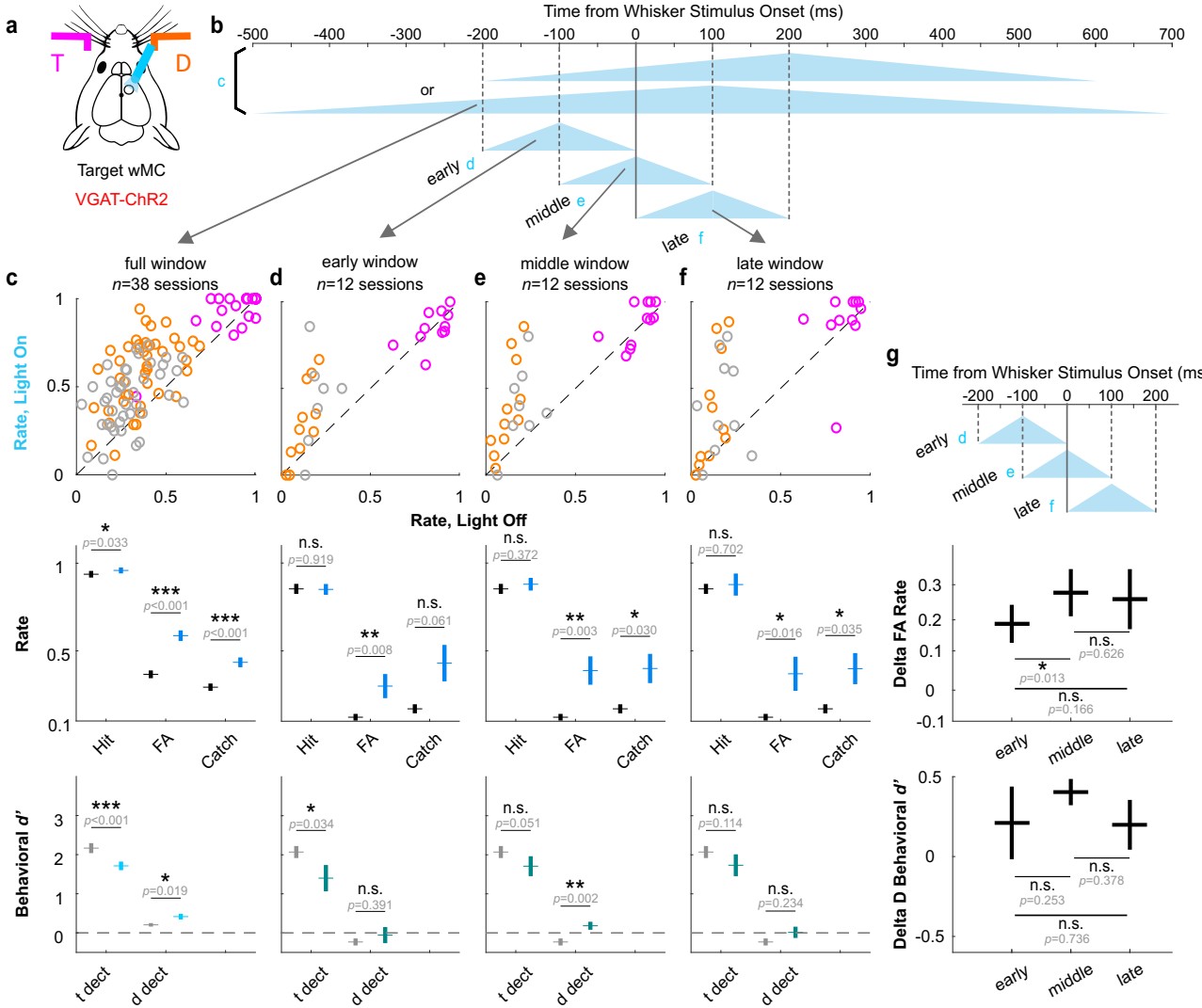

**Fig. 6 | Effects of suppressing target-aligned wMC in transient, varying time windows. a** Illustration of suppressing target-aligned wMC in VGAT-ChR2 transgenic mice. **b** Illustration of different profiles of optogenetic light. "**c**" indicates the light profiles used the previously described experiments. "**d**", "**e**", and "**f**" indicate the light profiles used for transient varying time wMC suppression. **c** Behavioral results of light profile "**c**", as same as Fig. 1g left row, and included here for reference. **d**–**f** Behavioral results corresponding to light profiles "**d**", "**e**", "**f**", respectively. **g** Statistical comparison for the behavioral results in [**d**–**f**]. "Delta FA Rate" is calculated as light_on false alarm rate - light_off false alarm rate from the middle row of [**d**–**f**]. "Delta D Behavioral *d*"" is calculated as light_on behavioral *d*' - light_off behavioral *d*' of distractor detection from the bottom row of [**d**–**f**]. Sample sizes are shown in the figure. Statistical comparisons are from two-tailed paired t-tests, without correction for multiple comparisons. Data are represented as mean ± SEM.

not have the head-fixation confounds of stress[49] and sensory-motor prediction errors[50].

We speculate as to the cellular and circuit mechanisms underlying these sensory processing modulations and their impacts on behavioral responses. Our analyses of pre-stimulus spike rates were notable for a preferential wMC activation of distractor-aligned S1 inhibitory neurons (Fig. 5g). While this increased inhibitory drive did not substantially suppress distractor stimulus encoding in distractor-aligned S1, we hypothesize that these inhibitory neurons may prevent the spread of distractor evoked responses into target-aligned cortex. Additionally, we have considered why the propagation of distractor responses into target-aligned S1 would impair distractor response suppression. In this selective detection task, we suspect that behavioral responses are conditioned on the activation of target-aligned regions, including from target-aligned S1. However, in naive subjects, whisker-evoked responses propagate widely and across hemispheres (Supplementary Fig. 5, see also refs. 35, 51–53). We reason that it would be difficult for a downstream reader to distinguish between target-aligned S1

activations that propagated bottom-up from target stimuli vs target-aligned S1 activations that propagated laterally from distractor stimuli. Thus, an important function of top-down signals would be to restrict the propagation of distractor stimuli into target-aligned S1, thereby increasing the likelihood (posterior probability) that target-aligned S1 activations reflect the occurrence of target stimuli. Without top-down restrictions of distractor propagation, downstream readers would be more likely to respond incorrectly to target-aligned S1 activations evoked by distractor stimuli (as simulated in Supplementary Fig. 12).

We note that wMC contributes to multiple behavioral measures in this task and recognize that modulations in S1 are likely to be only part of its function. wMC suppression substantially increased catch rates and, relatedly, increased signal detection theory measures of the tendency to respond (reduced the criterion). General reductions in the tendency to respond may be mediated through S1, or through other wMC projections, such as to the dorsolateral striatum[27], subthalamic nucleus[54], thalamus[55,56], and/or brainstem nuclei[57]. Further experiments determining behavioral changes from pathway-specific

perturbations are needed to resolve such contributions. Additionally, we recognize that in expert mice wMC does not appear to play a major role in increased target stimulus cross-hemispheric propagation (Fig. 3f, Supplementary Fig. 5e, f). Thus, we suspect the involvement of additional top-down neuromodulation[58–60] and/or bottom-up plasticity through training[61,62].

When are top-down cortical inputs activated to modulate sensory processing? A widely accepted framework is that stimulus-triggered bottom-up inputs are required to activate higher order regions, which in turn deliver feedback signals[26,39–41]. In this "reactive" framework, the bottom-up and top-down signals in sensory cortex occur sequentially and are temporally dissociable[26,63–66]. Additionally, this framework provides little motivation for studying the impacts of top-down regions on pre-stimulus activity. In contrast, we provide three lines of evidence suggesting proactive (pre-stimulus) wMC modulation of S1. First, differences in target stimulus and distractor stimulus cross-hemispheric propagation were evident remarkably early post-stimulus (statistically significant by 20 ms, Fig. 2g) which is typically associated with the initial feedforward sensory sweep preceding reactive feedback. Second, we causally demonstrated that wMC robustly modulates S1 before stimulus onset (see also[31]). Importantly, this pre-stimulus modulation is context-dependent, with a stronger drive onto putative inhibitory neurons in distractor-aligned S1 (Fig. 5g). Third, we demonstrated that inhibiting wMC for a short epoch at the time of whisker stimulus onset was sufficient to increase distractor detection (Fig. 6). Based on these observations we cannot rule out rapid reactive feedback mechanism. However, we propose that these findings are more consistent with a "proactive" framework, in which top-down regions set the initial conditions for sensory processing according to goal-direction and internal state well before a stimulus arrives[42].

## Methods
All experimental protocols were approved by the Institutional Animal Care and Use Committee of University of California, Riverside. Both male and female wild type (C57BL/6) (2 male and 9 female) and transgenic (VGAT-ChR2) (7 male and 6 female) mice were purchased from Jackson Laboratories (014548, 000664) and were subsequently housed and bred in an onsite vivarium. Sex differences were not analyzed due to insufficient statistical power. The mice were kept in a 12/12 h light/dark cycle, and the experiments were conducted during the light phase. At the beginning of the experiments, all mice were between 6-18 weeks old (age: mean±std: $82 \pm 26$ days). For information on all mice used in these experiments, see Supplementary Table 1.

### Surgery
Mice were anesthetized using an induction of ketamine (100 mg/kg) and xylazine (10 mg/kg) and maintained under isoflurane (1–2% in $O_2$) anesthesia. A lightweight metal headpost was attached to the exposed the skull over dorsal cortex, creating an $8 \times 8$ mm central window. Mice were treated with meloxicam (0.3 mg/kg) and enrofloxacin (5 mg/kg) on the day of the surgery and for two additional days after the surgery. For mice that were used for anesthetized recordings, the recordings were conducted immediately after surgery. For mice that were trained in the behavioral task, water restriction was initiated after recovery from surgery for a minimum of 3 days. For mice prepared for optogenetic experiments, the bone over the bilateral S1 and wMC was thinned under isoflurane anesthesia 2–3 days prior to the optogenetic perturbation. Between recordings, the skull was protected with Kwik-Cast (World Precision Instruments). For further surgical details, see[7,44],

### Whisker selective detection behavior and whisker stimulation
MATLAB software and Arduino boards were used to control the behavioral task. Head-fixed mice were situated in the setup during behavioral sessions. Two piezoelectric benders with attached paddles were used: one placed within one whisker field and assigned as the target; the other placed within the opposite whisker field and assigned as distractor. Both paddles targeted the D2/E2-D3/E3 whiskers within their respective whisker fields and the assignments of target and distractor remained consistent for each mouse throughout training and recording. Each whisker stimulus was a single rapid deflection, with a triangular waveform of fixed velocity and variable duration/amplitude. In each recording session, two stimulus amplitudes were applied: the large amplitude was near the saturation of the psychometric curve and the small amplitude was $1/4^{th}$ of the duration/amplitude of the large amplitude stimulus, within the dynamic psychometric range. The amplitudes were customized for each mouse, in expert mice ranging from 1.4 ms to 11.2 ms duration. In all behavioral sessions, target and distractor amplitudes were equivalent. The number of target vs distractor trials and large amplitude vs small amplitude trials were approximately balanced and pseudo-randomly presented. Additionally, 20% of non-reward catch trials without whisker stimuli were distributed pseudo-randomly throughout the session.

A 200 ms lockout period (delay) was introduced after whisker stimulus onset and before the response window. Licking during the lockout period was punished by aborting the current trial and resetting the inter-trial interval (6–10.5 s). After the lockout period, there was a 1-s response window. Licking during the response window of a target trial was considered a "hit" and was rewarded with a fluid reward (~5 μl of water) from the central lickport. All other licking was punished with resetting the inter-trial interval. No licking during the response window of a distractor trial or catch trial was considered a "correct rejection" or "correct withholding", respectively, and was rewarded with a shortened inter-trial interval (1.4–3.1 s) and subsequent target trial. All mice were allowed to continue in the task until unmotivated, defined as >2 min of no licking and >3 min of no "hit" trials. Depending on mouse size, training stage, reward history, and likely other factors, a full daily session could vary from 200–400 trials occurring over 1-2 h. In post-processing of this raw session data, we set strict criteria for deciding which segment to use for further analyses (see "Data inclusion criteria and quality control").

Mouse weights were maintained above 85% of their pre-surgery weights throughout the training and recording sessions. Mice were considered as experts once their discrimination of target and distractor stimuli (discrimination $d' = Zhit\ rate - Zfalse\ alarm\ rate$, in which Z is the inverse of the normal cumulative distribution function) reached a threshold (discrimination $d'>1$) for three consecutive days. For all behavioral-physiological data presented in this study, mice were performing at expert level during data collection. For more training details and learning trajectories, see ref. 7; for more training setup information, see ref. 44.

### Whisker independent habituated behavior
Mice were prepared for head-fixed behavior as described above. However, during behavioral training, whisker stimuli were not associated with water rewards (which were delivered automatically) and mice were not punished for spontaneous licking. Trials consisted of water reward trials (30%-50% of all trials) and whisker stimulus trials (50%-70% of all trials) randomly interleaved during a session, with inter-trial intervals distributed from 6–12 s. Mice were habituated to this task for 7 days before initiating neuronal recordings. Task-engaged behavioral sessions typically lasted for 1.5 h and generated ~300 whisker stimulus trials. Whisker stimulus durations ranged from 2.8 ms to 11.2 ms, similar to expertly performing mice.

### Electrophysiological recording and whisker receptive field mapping
All electrophysiological recordings were obtained from 12 VGAT-ChR2 mice and 3 wild type mice. On average, 7 sessions were recorded from each mouse (range 1–18), and 175 units were recorded from each mouse (range 24–406) (Supplementary Table 1). These data include

both S1 recordings and wMC recordings (wMC recordings are described in "Optogenetic calibration" below). For electrode implantation in S1, a small craniotomy with durotomy (~0.5 mm in diameter) above the S1 barrel field (from bregma: 2.5–4 mm lateral, 1–2.2 mm posterior) was made under isoflurane anesthesia the same day of recording (used for up to 3 days of acute recordings). After ~30 min post-surgery, mice were tested in the behavioral task without electrode implantation to ensure recovery to normal behavior. Upon evidence of normal expert behavior, a silicon probe (Neuronexus A1x16-Poly2-5 mm-50s-177) was advanced into the brain using a Narishige micro-manipulator under stereoscope guidance. This silicon probe design has 16 electrode sites, arranged in two columns spanning 375 μm, allowing for tetrode clustering of single units. We positioned the recording sites to target layer 5, from 500 to 1000 μm below the pial surface (mid-point of the silicon probe recording sites: mean ± std, 620 ± 90 μm (Supplementary Fig. 7b)), 92 recording sessions in total. Whisker alignment for S1 recordings was verified by two methods (as also described in ref. [44]). First, during electrode implantation we verified correct alignment by hand mapping of several individual whiskers and observing real-time local field potential (LFP) display. Second, post-hoc we plotted the large amplitude stimulus evoked multi-unit spiking activity and only included sessions with robust and fast-rising peaks in the post-stimulus time window (peak response reaching at least 1.4× baseline activity within a 50 ms post-stimulus window).

### Optogenetic suppression and opto-tagging

To transiently suppress specific cortical regions, we used optogenetic stimulation of GABAergic interneurons in VGAT-ChR2 mice[67,68]. Control studies with identical optical illumination were performed in wild type mice without ChR2 expression. For optogenetic stimulation, LED fiber optic terminals were placed above wMC (centered on [from bregma]: 1 mm lateral, 1 mm posterior) and/or S1 (same coordinates as above). Optogenetic GABAergic stimulation was used for both suppression studies as well as for opto-tagging of recorded GABAergic interneurons. For both applications, 470 nm blue light was delivered through a 400 μm core diameter optical fiber (Thorlabs). Optogenetic stimulation was delivered using a triangular function, beginning 200–500 ms before whisker stimulus onset, peaking 100–200 ms after whisker stimulus, and decaying for 400–600 ms. This protocol was used to (1) prevent transient onset/offset effects of square pulse illumination and (2) ensure a relatively strong and stable suppression for 100 ms pre-stimulus through 100 ms post-stimulus. Maximum intensity at peak optogenetic stimulation was 18 mW/mm$^2$ (2.26 mW) at the fiber tip. For the behavioral sessions without electrophysiological recordings, optical suppression was applied on one third of all trial types, randomly distributed. For the sessions (awake or under anesthesia) with electrophysiological recordings, simultaneous optogenetic suppression and opto-tagging were conducted, in which optogenetic suppression was randomly applied on one third of all trial types and opto-tagging was randomly applied on 7–10% of all trial types.

Barrier and masking methods were used to prevent visual detection of the optical stimulus. Barrier methods consisted of wrapping the headpost and optical fiber in opaque material. Masking consisted of applying a 10 Hz blue light LED stimulation (duty ratio = 50%) directly above the subject's eyes on all trials, beginning 180 ms before opto-stimulation onset and ending 170 ms after opto-stimulation offset. Masking light was given on all trials (irrespective of optogenetic suppression or whisker stimulus type).

For a subset of the optogenetic behavioral sessions (presented in Fig. 6 and Supplementary Fig. 13) opto-stimulation was even further restricted to three 200 ms peri-whisker stimulus windows. The timing of these windows: early, beginning 200 ms before whisker stimulus onset and ending at whisker stimulus onset; middle, beginning 100 ms before whisker stimulus onset and ending 100 ms after whisker

stimulus onset; late, beginning at whisker stimulus onset and ending 200 ms after whisker stimulus onset. Each opto-trial type was delivered on 11% of trials, randomly interleaved along with control (non-opto) trials. All other optogenetic parameters were the same as described above. To increase statistical power, only large amplitude whisker stimuli were used. In these sessions, the masking light was delivered continuously (not trial-based).

### Optogenetic calibration

Two VGAT-ChR2 mice were used for optogenetic calibration of wMC (1 mm lateral, 1 mm anterior from bregma) optogenetic suppression. Target-aligned wMC was recorded in awake mice during performance of the whisker selective detection task (7 recordings sessions total). The tip of the recording probe ranged from 800 to 900 μm below pial surface (mid-point of the silicon probe recording sites: mean±std, 620 ± 40 μm). In each session, we tested both direct suppression (light positioned above target-aligned wMC) and indirect suppression (light positioned above distractor-aligned wMC) in block design. Opto-stimulation parameters were the same as described in "Optogenetic suppression and opto-tagging" above.

### Electrophysiological recording data pre-processing

Neuralynx software was used for data acquisition and spike sorting. Putative spikes were identified as threshold crossings over 18–40 μV. Spike sorting and clustering were conducted offline by SpikeSort3D software, first through the KlustaKwik function followed by manual inspection of waveform and inter-spike interval distributions. On average, 23 units were identified in each recording session (mean±std, 23.4 ± 5.6 units). Further data analyses were conducted using MATLAB software (MathWorks). Spike times were binned within 5 ms non-overlapping bins. For more details about spike sorting, see[44].

### Whisker movement imaging and data preprocessing

To acquire the videos, we used a CMOS-sensor camera (Thorlabs CS165MU1/M - Zelux with Edmund Optics lens 33301) with a band pass filter (Edmund 89829, center wavelength: 660 nm, full width half max: 66 nm) to eliminate the masking and optogenetic light. The whisker fields were constantly illuminated with a red LED. The whisker imaging camera was positioned directly above the mouse. Field of views included both paddles and both whisker fields (including whiskers that were and were not contacted by the paddles). The videos were acquired within a trial structure, starting from 1 s before whisker stimulus onset and continuing through the whisker stimulus and response windows, at a frame rate of 35.6 Hz (during the selective detection task) or 58.82–66.67 Hz (during the whisker independent habituated behavior). In post-processing, four regions of interest (ROIs) were manually selected: target paddle, distractor paddle, target-aligned whiskers, distractor-aligned whisker. Only whiskers not touching the paddles were selected, to visualize whisker movements not obstructed by the paddle. The MATLAB function vision.-VideoFileReader was used to exact video information. Whisker motion energy was calculated as the summed, normalized (squared) difference between subsequent frames of single pixel gray values across a ROI. Temporal alignments between whisker imaging and neuronal recordings were verified by paddle motion energy.

### Widefield Ca$^{2+}$-sensor imaging

Neuronal activity of dorsal cortex was imaged in GCaMP6s expressing mice during expert performance in the selective detection task. The imaging datasets analyzed here were previously published[7,69]. The dataset consists of $n = 40$ behavioral-imaging sessions from $n = 5$ mice. Analyses included all target trials and all distractor trials, regardless of behavioral outcome. Imaging data were processed for pixel-by-pixel whisker stimulus encoding (neuronal $d'$), comparing pre-stimulus (stimulus absent) and post-stimulus (stimulus present) time points[7].

The analyses presented in this study are from the 100–200 ms post-stimulus imaging frame, during the lockout period and before the response window. Regions of interest (ROI) were selected to include primary somatosensory cortex, including the dorsomedial portion of the S1 whisker representation. Analyses focused on whisker stimulus encoding in the unpreferred hemisphere (ipsilateral to the whisker stimulus). $d'$ values were averaged within each ROI and compared for target and distractor stimulus trials.

### Anterograde neuroanatomical tracing and histology

AAV1-CamKIIa-halo-YFP virus particles (Addgene, 26971, 300 nL per mouse) were injected into unilateral wMC in 5 mice (C58BL/6 J wild type). After 4 weeks of expression, mice were perfused by 4% paraformaldehyde (PFA). The brains were sliced by a vibratome (Leica VT1000 S) for 100 μm coronal sections. Fluorescence microscope Keyence BZ-X710 was used for imaging. For each mouse, one section with clear S1 barrel structure and robust YFP expression was selected for laminar projection analyses.

### Data analysis

All data were analyzed using SPSS or custom-written MATLAB scripts. ANOVA was conducted in SPSS by univariate general linear model. Data are reported as mean ± SEM, unless otherwise stated.

**Behavior analyses.** For target detection, behavioral $d' = Z_{Hit\ Rate} - Z_{Catch\ Rate}$. For distractor detection, behavioral $d' = Z_{False\ Alarm\ Rate} - Z_{Catch\ Rate}$. For target-distractor discrimination, discrimination $d' = Z_{Hit\ Rate} - Z_{False\ Alarm\ Rate}$. For target responding criterion, $c = -0.5 \times (Z_{Hit\ Rate} + Z_{Catch\ Rate})$. For distractor responding criterion, $c = -0.5 \times (Z_{False\ Alarm} + Z_{Catch\ Rate})$. Z is the inverse of the normal cumulative distribution function. To avoid yielding infinite values, the "log-linear rule" was used for all trial types, adding 1 additional trial to each trial type, split equally (0.5) into responding and non-responding outcomes[70].

**Neuronal data analyses.** Neuronal encoding $d'$ across a 100 ms post whisker stimulus window: $d' = \sqrt{2} \times Z_{AU\text{-}ROC}$. AU-ROC is the area under the receiver operating curve (ROC), in which the curve is composed of the spike counts within 100 ms pre-stimulus windows and 100 ms post-stimulus windows. Each window was composed to two 50 ms epochs. Neuronal $d'$ across 20 ms post stimulus sliding windows: $d' = \sqrt{2} \times Z_{AU\text{-}ROC}$. This calculation is similar to that described above, except the post stimulus window was a single 20 ms window and was compared to 5 consecutive 20 ms pre-stimulus windows. The post-stimulus 20 ms sliding window was sampled every 5 ms, thereby consisting of 75% overlap, up to 100 ms post-stimulus. To account for the 5x sampling of the pre-stimulus window, we multiplied the post-stimulus histogram by 5 before plotting the ROC.

**Whisker imaging analyses.** For post-stimulus analyses, for each trial the whisker motion energy on post-stimulus frames were normalized by subtracting the whisker motion energy from the immediately pre-stimulus frame. When comparing between whisker fields, whisker motion energy distributions per session were additionally z-score transformed to account for differences in imaging (camera position, lighting) and windowing. To ensure that differences in post-stimulus whisker motion were not obscured by z-scoring, we determined the session μ and σ from pre-stimulus data only.

**Neurometic-psychometric correlation analyses.** For target-aligned S1 recording sessions, we analyzed the session-by-session correlations between changes in distractor stimulus encoding and changes in behavioral measures due to target-aligned wMC suppression. Four behavioral outcomes were analyzed: false alarm rate, catch rate, distractor detection, and distractor criterion. Changes in distractor stimulus encoding for each session was calculated as the difference of averaged neuronal $d'$ across the 30–100 ms window after distractor stimulus onset between light-on and light-off trials.

**Linear fitting and confidence interval analyses.** For the linear fitting of the single unit population data and to determine the slopes with 95% confidence intervals, we used MATLAB functions: the polyfit function was used to calculate the slope for the linear fit, and the polyparci function was used to calculate the 95% confidence intervals. These data were plotted by the fitlm function.

**Classification analyses.** We trained a linear discriminant model (MATLAB function: fitcdiscr, with diagonal covariance matrix) to classify the spike rates of target-aligned S1 neurons into target or distractor whisker stimulus trial types. Out of all the putative excitatory units recorded from target-aligned S1 ($n = 197$), the half of units with largest target whisker stimulus neuronal $d'$ ($n = 98$) were pooled together as pseudo-ensembles. Spike rates on each trial were determined as the increase in spiking from pre-stimulus baseline, using only the first 50 ms post-stimulus to better isolate the initial sensory response. To equalize the number of trials across sessions, we reconstructed the data: the sessions with less trials had their trials duplicated and appended to the original trials to match the trial number of the session with the most trials; for the sessions with the trial numbers not a common divisor of the trial number in the longest session, the trials were randomly sampled (without replacement) from these sessions accordingly and added to that session to fill in. After data reconstructions, each unit contained 70 target or distractor light-off trials and 30 target or distractor light-on trials. For each simulation, the trial order for all the units were randomly shuffled to remove the temporal structure and correlation between units. 40 target whisker stimulus light-off trials and 40 distractor whisker stimulus light-off trials were used to train the linear discriminant model. We then tested the model on the 30 light-off hold-out trials and the 30 light-on wMC suppression trials. This process was iterated 1000 times. Data are presented as mean ± standard deviation.

**wMC→S1 axonal histology analyses.** ImageJ was used for ROI selection, spanning (width) 400–1000 μm and (depth) all layers of neocortex. Fluorescence intensity distributions averaged across ROI width were extracted by "Plot Profile" for each mouse, and data were averaged across mice in MATLAB. For data presentation, cortical depth of the histological profile was multiplied by 1.2 to adjust for brain tissue shrinkage during fixation.

### Data inclusion criteria and quality control

"Task engaged" was defined as a continuous period of at least 10 min with no greater than a 60 s gap of non-licking. For any session with more than one engaged period, only the longest continuous segment was used for further analyses. Sessions without continuous engagement for at least 10 min were excluded from further analysis. "Task disengaged" epochs always followed periods of task engagement. Disengaged epochs were recorded for ~30 min and contained at least 300 trials. We suspect that task disengagement was due to lack of motivation from satiety after receiving sufficient rewards. For analyses of behavioral outcomes, at least 5 trials of a given type were required to be included in the session analysis. For trial numbers for all the analyses, see Supplementary Table 2.

For recording data during selective detection task performance, we analyzed only the "task engaged" epochs as defined above, so that the behavioral and neuronal data were based on the same trials. In Fig. 5b, "VGAT" is based on all sessions recorded from VGAT-ChR2 mice, regardless of the engagement status or wakefulness. Similarly, Supplementary Fig. 10 is based on all recording sessions.

For whisker independent habituated behavior, whisker movements were imaged and analyzed for all the whisker stimulus trials. In the analyses presented, we excluded the half of trials with highest whisker motion energy within 500 ms before stimulus onset, since ongoing whisking during stimulus presentation reduces S1 sensory-evoked responses[71].

### Principal whisker alignment inclusion criteria

Out of 78 recording sessions, 14 were unaligned, 64 were aligned, based on criteria described above. Only the aligned sessions are used for neuronal data analyses. wMC suppression sessions with unaligned S1 recordings were included for behavior only analyses, along with wMC suppression sessions without neuronal recordings.

### Visual mask-induced noise filtering

For optogenetic sessions, we applied a 10 Hz mask light above the eyes of mice, which sometimes caused a 10 Hz (onset triggered) or 20 Hz (onset and offset triggered) signal in the neuronal recording data (likely due to visual transduction). To minimize the influence of this contamination, we applied a band-pass filters ("designfilt" function of MATLAB, band-stop finite impulse response filter, designed based on a two-way Butterworth notch filter, with low-high cutoff frequencies of 9–11 Hz and 19–21 Hz, filter order = 2) to the spike count time series for single unit or multi-unit data before performing subsequent analyses. Supplementary Fig. 14 shows key neuronal analyses with and without filtering.

### Putative excitatory and putative inhibitory neurons inclusion criteria

To identify putative excitatory or inhibitory units, we analyzed the time window of the optogenetic light duration (excluding the first and last 100 ms time bins at the start and the end of the triangle waveform) to calculate the averaged trial-by-trial light-on spike rate for each unit (using an equivalent window for light-off control trails). We calculated the difference of averaged rates between light-on trials and light-off trials and determined significance from two-sample $t$-tests with a threshold of $p < 0.1$. Out of these significantly modulated units, if the average spike rate of light-on trials was larger than that of light-off trials (enhanced), the units were considered as putative inhibitory neurons; if the average spike rate of light-on trials was smaller than that of light-off trials (suppressed), the units were considered as putative excitatory neurons. Any unit with $p \geq 0.1$ was not categorized and not included in further cell-type analyses.

### Reporting summary

Further information on research design is available in the Nature Portfolio Reporting Summary linked to this article.

## Data availability

All data used and reported in this study have been deposited in the Figshare database including a "Source Data" file under accession code: https://figshare.com/articles/dataset/Zhang_Zagha_code_data_NatureComm/22191193. Additional data that support the findings of this study are available from the corresponding author E.Z. upon request. Source data are provided with this paper.

## Code availability

Custom-written code used to analyze data is available at Figshare (https://figshare.com/articles/dataset/Zhang_Zagha_code_data_NatureComm/22191193).

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

## Acknowledgements

We thank the members of the Zagha lab for discussions regarding experimental design and data analyses. We especially thank Krithiga Aruljothi and Dr. Krista Marrero for providing the widefield imaging dataset. We also thank Dr. Behzad Zareian for developing the whisker imaging pre-processing methods. We thank Dr. Hongdian Yang and Dr. Chunyu Ann Duan for their valuable comments on a previous version of this manuscript. This research project was supported by NIH/NINDS R01 Grant NS107599, E.Z., and Whitehall Foundation Grant 2017-05-71, E.Z.

## Author contributions

Z.Z. and E.Z. initiated the project and designed the experiments. Z.Z. performed the experiments and acquired the data. Z.Z and E.Z. analyzed the data and prepared the manuscript.

## Competing interests

The authors declare no competing interests.
