## [Peer Review File · Nature Communications]

Motor Cortex Gates Distractor Stimulus Encoding in Sensory CortexREVIEWER COMMENTS

Reviewer #1 (Remarks to the Author):

In this manuscript, Zhang and Zaghera used a whisker-based target-distractor Go/NoGo task to examine how the whisker region of motor cortex (wMC) modulates the responses in primary sensory cortex (S1). The authors found that encoding of distractor stimulus in target-aligned S1 was suppressed, and inhibition of target-aligned wMC caused an increase in the encoding of distractor stimulus and a decrease in stimulus selectivity in target-aligned S1. The authors also found that wMC inhibition during a pre-stimulus period reduced spike rates of S1 neurons, supporting a framework of proactive modulation. The results in this manuscript provide important insights into the function and mechanism of top-down feedback, and will be of importance in the field.

The paper can be further improved before it is suitable for publication in Nature Communications.

Major:

1. In Fig. 2E and 2G, the authors analyzed the encoding of target or distractor stimulus in target-aligned S1 or distractor-aligned S1. The encoding of distractor stimulus in target-aligned S1 was suppressed, whereas the encoding of target stimulus propagated to distractor-aligned S1. Is the propagation of target stimulus across hemispheres a result of behavioral training? Does the suppressed encoding of distractor stimulus in target-aligned S1 depend on trial outcome (FA vs. CR) in addition to task engagement?
2. The authors demonstrated that target-aligned wMC suppression increased the encoding of distractor stimulus in target-aligned S1 (Fig. 3), and suggested that the increased encoding of distractor stimulus in target-aligned S1 may contribute to the increase in FA rate during wMC suppression (lines 265 – 266). The authors could test this conjecture by examining whether the increase in FA rate correlates with the increase in d prime of distractor stimulus in target-aligned S1.
3. In the discussion, the authors hypothesized that the pre-stimulus activity of inhibitory neurons in distractor-aligned S1 may prevent the spread of distractor evoked responses into target-aligned S1. During light off trials in Fig. 5E, is the pre-stimulus spike rate of inhibitory neurons indeed higher in distractor-aligned S1 than in target-aligned S1, or the pre-stimulus spike rate of excitatory neurons lower in distractor-aligned S1 than in target-aligned S1?
4. Given that wMC modulation of S1 pre-stimulus activity occurred under anesthesia, one would like to know whether such proactive modulation causally contributes to behavioral performance. To directly test the role of wMC proactive modulation and the functional importance of distractor-aligned S1 inhibitory neurons, an interesting experiment would be to manipulate the pre-stimulus activity of these inhibitory neurons and examine the impact on behavioral performance and encoding of distractor stimulus in target-aligned S1. As additional experiments are required, this is up to the authors.

Minor:

1. In line 628, the authors reported that 7 sessions were recorded from each mouse (range 1-18). It would be helpful to clarify on average how many neurons were recorded from each mouse.
2. In Fig. 1 and Fig. S1, the authors showed that suppression of distractor-aligned wMC or distractor-aligned S1 both increase FA rate. However, suppression of distractor-aligned wMC did not affect stimulus encoding in distractor-aligned S1. The authors should probably explain whether suppression of distractor-aligned wMC influences the stimulus encoding in target-aligned S1.
3. The authors recorded from Layer 5 S1 neurons. It would be helpful to show the laminar distribution of wMC axon terminals in S1.
4. For Fig. 4 and Fig. 5, does 'wMC suppression' mean 'suppression of target-aligned wMC'?

5. Line 561, 'casually' should be 'causally'.

Reviewer #2 (Remarks to the Author):

Review comments for Zhaoran Zhang and Zaghera, 2022-07.

This study examines how the whisker region of motor cortex (wMC) modulates somatosensory cortex (S1) activity and contributes to sensory detection. The authors trained head-fixed mice on a go/no-go task to detect a whisker deflection on one side (defined as the target stimulus) and report by licking to a lickport, but to ignore the whisker deflection on the other side of the whisker field (defined as the distractor stimulus). The authors found that optogenetic suppression of the wMC on either hemisphere increased the false alarm rate. At the population level, suppression of wMC contralateral to the target stimulus increased the coding for distractor by S1 ipsilateral to the distractor. At the single unit level, suppression of the same wMC increased the correlation between S1 coding for target and distractor. Finally, the authors noticed that the optogenetic suppression of wMC activity during the time epoch preceding stimulus onset also influenced S1 activity with higher preference to putative inhibitory neurons.

Overall, this is an interesting study addressing an important problem of how top-down feedback input contributes to target-distractor discrimination during sensory detection. Technique wise, it is a plus to use head-fixed and well-trained behavior combined with optogenetic perturbation and cell-type specific single unit recording. The data analyses are sound and the results from different experiments are generally consistent with each other. However, there are several important drawbacks in this study. The observations are mostly at the phenomenological level, with limited mechanistic insights. While the briefly mentioned change in the E-I ratio following wM1 silencing can be helpful, the understanding of how wM1 suppresses distractor information is still limited. In a bigger picture regarding the sensorimotor cortical circuits, there is important information missing, and some of the results in this study are difficult to understand and even puzzling. In addition, some of the general conclusions need to be tailored based on the experimental results.

Specific points

1. Some of the general conceptual claims need to be clarified. This study focuses on the interaction between wMC and S1 barrel field. The wMC is a subregion of the primary motor cortex (M1), usually termed as wM1, or vM1 (vibrissal M1). In rodents, the wM1 is strongly and reciprocally connected with S1 barrel field, forming an active sensing system. It is questionable to what degree that the wM1-S1 interaction represents a general frontal top-down mechanism for distractor suppression. The term 'frontal cortex' that the authors repeatedly mentioned usually refers to a wide range of areas ranging from many association cortices, prefrontal areas, and higher-order motor cortices. It is debatable and not widely accepted that wM1 belongs to this category of cortical regions. How representative is the phenomenon that the authors observed for the wM1-S1 interaction regarding the frontal cortical modulation of sensory processing? It is thus an overreaching conclusion, not strongly supported by the current results, for the following statement in the abstract, "In contrast to current models of frontal cortex function, frontal cortex did not substantially modulate the response amplitude of preferred stimuli. Rather, frontal cortex specifically suppressed the propagation of distractor stimulus responses, thereby preventing target-preferring neurons from being activated by distractor stimuli." 
Similarly, in line 263, the authors stated, "From these data, we conclude that the main impact of wMC on sensory encoding is to suppress the propagation of distractor stimulus responses in to target-aligned S1." There are certainly other important, and probably also 'main', impacts of M1 on sensory processing, e.g., as were demonstrated in Xu et al, 2012, Ranganathan et al, 2018. The phenomenon observed in the current study is not necessarily the main effect of M1 on S1.

2. A major drawback of this study is the lack of recording of wM1 activity and the related whisking

motion. It has been established that wM1 neurons are driven by both whisker sensory input and active movement of whiskers (Huber et al, 2012), and the activity in the wM1-S1 projections has been shown to encode whisker motion information (Petreanu et al, 2012). Moreover, the whisker system is an active sensing system, where sensory processing involves active movement of the sensors. Whisker stimulation on the two sides differentially defined as target and distractor and respectively associated with go and no go responses would certainly evoke distinct whisker movement patterns. Such differential whisker movements are likely to play an important role in the discrimination of target vs. distractor stimulus, and could be associated with different activity patterns in the wM1 in the two hemispheres. To understand how wM1 contributes to the sensory dependent go/no go behavior, it is important to monitor the whisker motion and record neuronal activity in wM1 during the task. Otherwise, some of the major results in this study stays at phenomenological level, and remain difficult to understand. For example, why suppression of wM1 in either hemisphere both led to increased false alarm rate (Figure 1)? How did the optogenetic suppression of wM1 affect whisking motion? Why suppression of wM1 contralateral to target stimulus marginally changed the activity of ipsilateral S1 while selectively increased the d-prime for the ipsilateral distractor (Figure 3)? Why suppression of wM1 contralateral to distractor did not have any effect? Could such asymmetrical effects be due to asymmetry in whisker motion and in wM1 activity between the two hemispheres?

3. The claim of proactive effect of wM1 on S1 is preliminary. It is certainly an important concept that some of the top-down inputs exert proactive or predictive effect. In the current study, the authors showed that the light stimulation before the whisker stimulus onset also had an effect on S1 pre-stimulus activity. However, the light stimulation covered both the pre-stimulus and post-stimulus periods. The authors did not examine whether the pre-stimulus suppression of wM1 only did indeed impact the encoding of either target or distractor stimulus in S1, or the behavioral performance. To claim that the wM1 proactively modulate sensory processing in S1, the authors should restrict the suppressing light stimulation of wM1 within the pre-stimulus time epoch, and examine whether there would be a similar effect on behavior as shown in Figure 1G, or on sensory coding as shown in Figure 3 and 4. Otherwise, the current results cannot distinguish the 'proactive' vs 'reactive' natures of the wM1-S1 modulation, and the last section and Figure 5 are not adding much value to this study.

Reviewer #3 (Remarks to the Author):

Introduction: The manuscript by Morishita and colleagues uses a combined approach of optogenetics, electrophysiology, and fiber photometry, with cognitive testing. Mice were assessed on a head-fixed go/no-go task, where they were trained to selectively response to whisker stimulation in one area and withhold responding in another. Optogenetic suppression of the whisker region of motor cortex (wMC) via GABAergic activation was assessed. Electrophysiological recordings of primary somatosensory cortex (S1) were assessed during normal task performance and in combination with wMC suppression. The authors conclude that wMC modulates S1 to primarily facilitate the processing and response inhibition towards non-target stimuli. Additionally, these data indicate towards a top-down modulation of S1 activity by wMC input, prior to the presentation of a stimulus.

The authors use a number of techniques, for the most part well-chosen, that offer a degree of temporal and cellular specificity. In particular, the use of in vivo optogenetics allows within-subjects analysis and causal manipulations, which the authors write have not been used previously to answer this particular question. Multiple controls are provided for the potential confounds of optogenetic stimulation. The paper provides a well-described assessment of the somatosensory cortex during this behavioral paradigm, which could be useful to researchers interested in these processes.

One limitation of the study is the specificity of the manipulations to the circuit under study, and the relationship with behavioral changes observed. The aim is to understand how wMC might modulate S1 and how that modulation affects behavior. However, the optogenetic suppression of wMC will suppress a number of circuits involving wMC – thalamus, striatum, and brainstem for example – and the suppression of any of these circuits could be the cause of any behavioral

changes observed. Therefore, the design of the experiment is not appropriate to determine the causal relationship between wMC suppression-induced S1 changes and behavior. The authors are aware of this, and indeed discuss how some behavioral changes are likely not due to changes in S1. However, it is not difficult, using optogenetics, to isolate the specific projections from wMC to S1; this is a strength of optogenetic approaches as opposed to, e.g., temporary inactivations produced pharmacologically. For me, the lack of specificity is a limitation that could be easily overcome and is necessary to draw the conclusions the authors wish to make.

Another question I have concerns the suppression of wMC by stimulating GABA-ergic interneurons. As far as I can tell, there was no electrophysiology done in wMC and therefore no assessment of to what extent this approach does suppress activity in (the pyramidal cells) of wMC. This is important, as there are reports that optogenetic stimulation of frontal cortical interneurons can in fact enhance cognitive function (and in tasks similar to that used by the authors).

A third limitation is that the animals under study are likely not normal. The head-fixed procedure is stressful and can drastically raise the level of stress hormones and alter behavior (see e.g., Juczewski et al., 2020, Scientific Reports). Studies using the head-fix method must at least clearly acknowledge this limitation.

I am also unclear what the authors are intending to study psychologically. On one hand, it seems that they wish to study attention: they refer to "detection" of the stimulus, and "distractors". However, to properly study stimulus detection, the detectability of the stimulus must be manipulated parametrically, and a relationship demonstrated between the detectability of the stimulus and the experimental manipulations by, for example, altering the stimulus duration or intensity of the stimulus. Distractors can be introduced by presenting distracting stimuli simultaneously with the target stimulus. Good examples of how to properly study attention are studies involving the 5-choice serial reaction time task, or the rodent continuous performance task. The set-up the authors use is probably best described as studying the well-learned performance of a simple go/no go task. In this context is the incorrect stimulus best described as a distractor, or is it simply a non-rewarded stimulus? This is important, as it changes the interpretation of the manipulations and the (possibly, see above) behavior-related activity in S1. Furthermore, depending on what they intend to study, alternative tasks – for example a task with symmetrical responses to control for movement confounds, addressed by the authors in the Discussion – might have been more appropriate.

The control animals may also be a limitation. It is stated only that they are wild type mice. Ideally, they would be identical to the experimental animals but without the expression of ChR2.

Furthermore, they must be littermate controls (many journals now require this). Additionally, the authors state that control studies in mice without optogenetic ChR2 expression were performed identically to the ChR2 expressing mice. Data from these mice do not appear to be included in Figure 1, where the optogenetic experiments are described. The authors mention electrophysiological recordings were taken from 2 wild-type mice (page 30), were these the only controls included for the optogenetic manipulations? The sex of the experimental mice was also not included, only that they were bred on site. What was the ratio of male to female mice? Were sex differences analyzed? Again, consideration of sex is rapidly becoming necessary for publication.

General comments:

- The introduction focuses on describing frontal cortex and its communication with sensory cortices to modulate response inhibition when faced with distracting (non-rewarded? See above) information. The authors then specify that they will be manipulating motor cortex and its pathway with the somatosensory cortex. Was this region chosen as a primary top down modulator of behavioral inhibition, or more so because of the nature of this task specifically? The role of this region specifically in the context of controlling response inhibition to distracting information is not discussed in the introduction.

- In the discussion, the authors focus on the involvement of these regions in distractor suppression, while some features of the data do not suggest this type of selectivity:
 - o 1. The data indicate that wMC suppression may increase both hit rate and false alarms, as well as premature responding. This is muddled by the discussion of small and large amplitude stimulation, as well as "depending on the statistical calculations used". The inclusion of these different amplitude stimulations appears under-realized, and it is unclear if there are any considerable differences resulting from using these two different protocols. The authors state that "unless otherwise stated, the data is reflecting large amplitude stimulation", so the purpose of including the small amplitude stimulation could be described more clearly.

- o 2. Suppression of wMC led to increases in target-aligned S1 neurons to both target and distractor stimuli.
 - o 3. Together, and the authors do mention this, the data here indicate that this circuit may be more specific to the gating of incoming stimuli and associated behavioral responses, whereas wMC suppression leads to more "liberal" assessment of stimuli and responding. However, the first part of the discussion is focused on the distractor/non-target side of this, and perhaps should be re-structured to reflect that these data do not appear specific to non-target information processing.
- Comments on the Methods section:
- There are a few questions about the task that I am hoping the authors could expand upon:
 - o Definition of expert: Is this term just being used as a description in this manuscript or is there a basis that having a d' over 1 is considered expert performance. Is 1 close to the upper range of scores that is possible for d' on this task? Ideally a statistical definition of expert-level performance would be given.
 - o Session criteria: What led to the termination of a session, if both the number of trials (200-400) and the time (1-2 hours) were variable? Were all animals subjected to an equal number of trials, or a minimum, for their data to be compared to each other?
 - o Task disengagement trials: It is mentioned that animals only needed to complete a minimum of 5 trials from either engagement type to be considered for analysis. Could the authors include a mention of how much variance there was across their cohort?
 - o Non-rewarded catch trials: On page 5, the authors state "catch trials (spontaneous response rate) when describing their measures. Could the authors please briefly elaborate why the catch trials are included and the purpose they uniquely serve?"
 - o Analysis for response measures = 0: "Response rates of 0 and 1 were adjusted as 0.01 and 0.99 respectively to avoid yielding infinite values."
 - Can the authors discuss why this correction method was used, or provide reference if this practice has been published before? Other methods correct for zero misses/false alarms using a formula to calculate a d' that take into account the potential possibility of one of these response types occurring in the future. For reference: <https://link.springer.com/article/10.3758/BF03203619>
 - The masking used to reduce confounds of the optogenetics resulted in noise in the 10hz and 20hz range. Can more details be included to describe the filtering that was done to remove this noise? Could removing neural activity in these ranges from the analysis be missing true neural activity, as in do the neurons being recorded from tend to fire in this frequency during normal task epochs?
 - When describing the calcium imaging data, the following is included, "The imaging datasets analyzed here were previously published (Aruljothi et al., 2020; Marrero et al., 2022). The dataset consists of n=40 behavioral imaging sessions from n=5 mice. The analyses presented in this study are from the 100-200 ms post-stimulus imaging frame, during the lockout period and before the response window."
 - o Can the authors confirm this specific analysis was not included in previous publications, or mention what is novel about the data as it is shown in this manuscript?
- Comments on the Result section:
- When discussing the wMC suppression on pages 5 and 6:
- "Suppression of target-aligned or distractor-aligned wMC caused trends towards increased small amplitude hit rates (with statistical significance depending on calculation method, Supplementary Fig. 1A,B)."
 - o Can the authors specify what statistics were used to assess the optogenetic effects and were they consistent for all measures?
 - S1 suppression: Could a c (response bias – signal detection theory calculated as $-0.5(z(HR) + z(FAR))$) measure be used with these data to characterize if the bi-directional effects of wMC and S1 suppression are generalized reduction or increases in response rates? (Rather than specific to hit rate or false alarm rates statistically.)
 - Modulation index characterization Figure 1G: The results for the modulation depict responses across all sessions and appear to include the combination of large vs. small amplitude trials.
 - o Can the authors confirm if this is the case, and if so, why these data were combined? Was statistical analysis included to confirm the small vs. large amplitude stimulation were not significantly different?
 - Figure 1 legend: "Optogenetic suppression (blue light-on) of wMC or S1 was performed on 33% of trials, randomly interleaved with control (light-off) trials."
 - o Can the authors elaborate why optogenetic stimulation occurred at this rate, as opposed to an

even split of trials with optogenetic stimulation vs. without?

When discussing electrophysiological recordings of S1 neurons on pages 8-10:

- Figure 2 legend: There is unclarity in the use of the term "d prime" as an index of neuronal activity due to the use of d prime as a primary behavior measure in the cognitive task. Similarly, using the term "stimulus" to describe both whisker stimulation and optogenetic stimulation. For both of these, it may be better to choose difference words to avoid confusion.

o It may be worthwhile to create a variation of this name or include the definition in the main text (as opposed to only in the figure legend).

o It may be ideal to keep the axes in d prime graphs E and G consistent (max 1.6 vs. max 1.2, respectively).

- "Neuronal activity during task disengagement was obtained from the same expert mice, after they stopped responding within a session (presumably due to satiety)."

o Could task disengagement result from a reduction in attention, opposed to simply motivation? Since the trial counts/time weren't the same for each animal, did some animals show prolonged engagement, or did engagement tend to fall off at similar time points during a session?

- "We found that target stimuli induced significant activation of distractor aligned S1 (n=40 sessions, d prime 0.14 ± 0.03 , one sample t-test $p=0.0002$). In contrast, distractor stimuli induced significant suppression of target-aligned S1 (n=40 sessions, d prime -0.07 ± 0.02 , one sample t-test $p=0.007$; paired t-test comparing target and distractor propagation, $p=1.8 \times 10^{-5}$ 207)."

o Can the authors confirm if the wording in the first sentence of this statement is correct? When looking at Figure S2, it appears that target stimuli induced activation of target aligned S1, not distractor aligned S1.

- "In marked contrast, disengagement led to an increase in distractor encoding in target-aligned S1".

o Could this finding be expanded upon in the discussion? As this was the only condition that increased during task disengagement, are there conclusions to be drawn from this finding? It is briefly mentioned on page 16, with the discussion that task disengagement led to reduced stimulus selectivity.

When discussing the effect of wMC suppression on S1 activity on pages 12-13

- "From these data, we conclude that the main impact of wMC on sensory encoding is to suppress the propagation of distractor stimulus responses into target-aligned S1." o Could the authors expand upon this conclusion? From the data, it appears that wMC suppression generally increased activity in target-aligned neurons regardless of stimulus type, as evident from increases in encoding target and distractor stimuli. Is this conclusion related to the differences in the time-windows that these populations showed increased activity?

When discussing the context-dependent nature of wMC modulation of S1

- "For recording conditions of anesthetized, awake behaving target-aligned, and awake behaving distractor-aligned, wMC robustly drives pre-stimulus spiking in S1 neurons."

o Could the authors expand on the animal's perception of the stimulus under anesthesia? How can pre-stimulus activity can be an index of top-down control of cognition in animals that are anesthetized? Surely this would indicate some kind of anticipatory period?

Reviewer #1 (Remarks to the Author):

In this manuscript, Zhang and Zaghera used a whisker-based target-distractor Go/NoGo task to examine how the whisker region of motor cortex (wMC) modulates the responses in primary sensory cortex (S1). The authors found that encoding of distractor stimulus in target-aligned S1 was suppressed, and inhibition of target-aligned wMC caused an increase in the encoding of distractor stimulus and a decrease in stimulus selectivity in target-aligned S1. The authors also found that wMC inhibition during a pre-stimulus period reduced spike rates of S1 neurons, supporting a framework of proactive modulation. The results in this manuscript provide important insights into the function and mechanism of top-down feedback, and will be of importance in the field.

The paper can be further improved before it is suitable for publication in Nature Communications.

Major:

1. In Fig. 2E and 2G, the authors analyzed the encoding of target or distractor stimulus in target-aligned S1 or distractor-aligned S1. The encoding of distractor stimulus in target-aligned S1 was suppressed, whereas the encoding of target stimulus propagated to distractor-aligned S1. Is the propagation of target stimulus across hemispheres a result of behavioral training? Does the suppressed encoding of distractor stimulus in target-aligned S1 depend on trial outcome (FA vs. CR) in addition to task engagement?

Thank you for your insightful comments.

Is the propagation of target stimulus across hemispheres a result of behavioral training? To address this question, we recorded S1 responses to preferred and unpreferred whisker stimuli in mice that were habituated to head-fixation but not performing a whisker detection task (Supplementary Fig. 5 C). Additionally, we compared the sensory-evoked cross-hemispheric (unpreferred) propagation from four conditions: expert target, expert distractor, naïve awake, and naïve anesthetized (Supplementary Fig. 5 E,F). We found that in both naïve awake and naïve anesthetized mice, there is modest cross-hemispheric propagation. This contrasts with expert mice which show robust propagation of target stimuli and suppressed propagation of distractor stimuli. Thus, behavioral training results in a bidirectional change in cross-hemispheric propagation. Notably, wMC suppression reversed distractor propagation suppression but not target propagation enhancement, suggesting that in expert mice wMC actively contributes to one aspect of this bidirectional modulation. These comments have been added to the manuscript.

Does the suppressed encoding of distractor stimulus in target-aligned S1 depend on trial outcome (FA vs. CR) in addition to task engagement? To answer this question, we compared target-aligned S1 responses on FA and CR trials (Fig. 2 H). Over the first 100 ms we did not observe a difference in responses to distractor stimuli according to trial outcome. However, when we extended the analyses into the response window

(Supplementary Fig. 4) we found that neuronal d' on FA trials was significantly larger than CR trials after approximately 350 ms post-stimulus. These findings are complementary to our previous findings of only late onset choice probability in target-aligned S1 for target stimuli (Zareian et al., 2021). This suggests that under normal, expert conditions, on a trial-by-trial basis of the same stimulus strength, CR/FA outcomes are not correlated with the variance of stimulus encoding in S1 but driven by variance in brain regions downstream of S1 (as also observed in non-human primate S1 in, e.g., de Lafuente and Romo 2006). Note, this does not mean that overall activity levels in S1 do not contribute to the decision to respond.

2. The authors demonstrated that target-aligned wMC suppression increased the encoding of distractor stimulus in target-aligned S1 (Fig. 3), and suggested that the increased encoding of distractor stimulus in target-aligned S1 may contribute to the increase in FA rate during wMC suppression (lines 265 – 266). The authors could test this conjecture by examining whether the increase in FA rate correlates with the increase in d' of distractor stimulus in target-aligned S1.

We performed the suggested analyses: correlating changes in FA rate with changes in target-aligned S1 encoding of distractor stimuli (Supplementary Fig. 8). We observed a non-statistically significant trend towards a positive correlation between these measures (slope=0.23, 95% confidence interval -0.20 to 0.65, $R^2=0.055$) (Supplementary Fig. 8 C). Furthermore, recognized (with the help of Reviewer 3) that changes in FA rates could result from changes in distractor detection and/or the tendency to respond. We used signal detection theory to transform changes in false alarm rates and catch rates into changes in detection (behavioral d') and tendency to respond (criterion) (summarized for all conditions in Supplementary Fig. 3). When we ran behavioral-neuronal correlations on these measures, we found a more consistent relationship between neuronal encoding and distractor detection ($R^2=0.12$) than the criterion ($R^2=0.0035$) (Supplementary Fig. 8 E,F). Given the vast subsampling of S1 neurons in our recording sessions we were surprised to find even these trending differences. Consequently, these findings support the more nuanced conclusion that wMC-mediated suppression of distractor stimulus propagation into target-aligned S1 specifically relates to the suppression of distractor detection.

3. In the discussion, the authors hypothesized that the pre-stimulus activity of inhibitory neurons in distractor-aligned S1 may prevent the spread of distractor evoked responses into target-aligned S1. During light off trials in Fig. 5E, is the pre-stimulus spike rate of inhibitory neurons indeed higher in distractor-aligned S1 than in target-aligned S1, or the pre-stimulus spike rate of excitatory neurons lower in distractor-aligned S1 than in target-aligned S1?

We analyzed the pre-stimulus spike rates of putative excitatory and inhibitory neurons in target-aligned and distractor-aligned S1. We found that indeed, as you suggested, the baseline activity of putative excitatory units is higher in target-aligned S1 than distractor-aligned S1 (which can be appreciated in Fig. 2 E and G, top row, from all units). This difference in excitatory neuron baseline activity is partially (~29%) due to the differential

drive from wMC, as determined by comparing differences in spike rates before and during wMC suppression. These analyses are included in the Results.

Despite the higher drive from wMC onto distractor-aligned S1 inhibitory neurons, their baseline firing rates were not higher than target-aligned S1 inhibitory neurons. We propose two possible explanations. First, this may reflect an inhibitory stabilized network, in which the excitatory drive onto inhibitory neurons is dominated by the local excitatory network (Tsodyks et al., 1997). An alternative explanation is that the main drive of wMC is onto a subset of inhibitory neurons which are not sufficiently resolved using the VGAT promoter (elaborated further in the next comment).

4. Given that wMC modulation of S1 pre-stimulus activity occurred under anesthesia, one would like to know whether such proactive modulation causally contributes to behavioral performance. To directly test the role of wMC proactive modulation and the functional importance of distractor-aligned S1 inhibitory neurons, an interesting experiment would be to manipulate the pre-stimulus activity of these inhibitory neurons and examine the impact on behavioral performance and encoding of distractor stimulus in target-aligned S1. As additional experiments are required, this is up to the authors.

We absolutely agree that this experiment would be highly revealing. However, before proceeding, we first want to identify the inhibitory neurons that are most strongly driven by wMC (by layer and cell-type). If we can identify a targetable interneuron sub-class, this would greatly enhance the specificity of our manipulation. We are currently preparing these exploratory experiments.

Minor:

1. In line 628, the authors reported that 7 sessions were recorded from each mouse (range 1-18). It would be helpful to clarify on average how many neurons were recorded from each mouse.

We now include Supplementary Table 1 indicating the behavioral-recording sessions and numbers of neurons recorded from each mouse.

2. In Fig. 1 and Fig. S1, the authors showed that suppression of distractor-aligned wMC or distractor-aligned S1 both increase FA rate. However, suppression of distractor-aligned wMC did not affect stimulus encoding in distractor-aligned S1. The authors should probably explain whether suppression of distractor-aligned wMC influences the stimulus encoding in target-aligned S1.

We conducted a related experiment that we believe can provide insight. Given the similarity of behavioral effects from suppressing target-aligned wMC and distractor-aligned wMC (as pointed out by Reviewer 2 comment 2), we wondered if this could be explained by their robust inter-hemispheric interactions (i.e., are we effectively suppressing both wMC regions when directly illuminating one region?). We experimentally addressed the specific question: what percentage of spiking in target-aligned wMC is suppressed by optogenetic inhibition of target-aligned wMC (direct) compared to optogenetic inhibition of distractor-aligned wMC (indirect)? As shown in

Supplementary Fig. 1, spike rate suppression of putative excitatory neurons in target-aligned wMC was 74% (direct) and 36% (indirect). Thus, to address your original comment, we would expect distractor-aligned wMC suppression to increase stimulus encoding in target-aligned S1, due to the robust coordination between wMC regions. These new findings and interpretations are described in the Results.

3. The authors recorded from Layer 5 S1 neurons. It would be helpful to show the laminar distribution of wMC axon terminals in S1.

We performed new axonal labeling experiments, showing the projections of wMC axons in S1 (Supplementary Fig. 15). These findings demonstrate robust wMC axon ramification in both layers 1 and 5, while sparing canonical thalamic input layer 4. Additionally, we reference the location of our S1 recordings to this histological analysis.

4. For Fig. 4 and Fig. 5, does 'wMC suppression' mean 'suppression of target-aligned wMC'?

For Fig. 4 A and Fig. 5 A, the label 'wMC suppression' refers generally to optogenetic suppression of wMC. Depending on the experiment, this could be target-aligned wMC, distractor-aligned wMC, or simply ipsilateral wMC (for anesthetized recordings). We now indicate this in the figure legends.

5. Line 561, 'casually' should be 'causally'.

Fixed.

Reviewer #2 (Remarks to the Author):

Review comments for Zhaoran Zhang and Zagha, 2022-07.

This study examines how the whisker region of motor cortex (wMC) modulates somatosensory cortex (S1) activity and contributes to sensory detection. The authors trained head-fixed mice on a go/no-go task to detect a whisker deflection on one side (defined as the target stimulus) and report by licking to a lickport, but to ignore the whisker deflection on the other side of the whisker field (defined as the distractor stimulus). The authors found that optogenetic suppression of the wMC on either hemisphere increased the false alarm rate. At the population level, suppression of wMC contralateral to the target stimulus increased the coding for distractor by S1 ipsilateral to the distractor. At the single unit level, suppression of the same wMC increased the correlation between S1 coding for target and distractor. Finally, the authors noticed that the optogenetic suppression of wMC activity during the time epoch preceding stimulus onset also influenced S1 activity with higher preference to putative inhibitory neurons.

Overall, this is an interesting study addressing an important problem of how top-down feedback input contributes to target-distractor discrimination during sensory detection. Technique wise, it is a plus to use head-fixed and well-trained behavior combined with

optogenetic perturbation and cell-type specific single unit recording. The data analyses are sound and the results from different experiments are generally consistent with each other. However, there are several important drawbacks in this study. The observations are mostly at the phenomenological level, with limited mechanistic insights. While the briefly mentioned change in the E-I ratio following wM1 silencing can be helpful, the understanding of how wM1 suppresses distractor information is still limited. In a bigger picture regarding the sensorimotor cortical circuits, there is important information missing, and some of the results in this study are difficult to understand and even puzzling. In addition, some of the general conclusions need to be tailored based on the experimental results.

Specific points

1. Some of the general conceptual claims need to be clarified. This study focuses on the interaction between wMC and S1 barrel field. The wMC is a subregion of the primary motor cortex (M1), usually termed as wM1, or vM1 (vibrissal M1). In rodents, the wM1 is strongly and reciprocally connected with S1 barrel field, forming an active sensing system. It is questionable to what degree that the wM1-S1 interaction represents a general frontal top-down mechanism for distractor suppression. The term ‘frontal cortex’ that the authors repeatedly mentioned usually refers to a wide range of areas ranging from many association cortices, prefrontal areas, and higher-order motor cortices. It is debatable and not widely accepted that wM1 belongs to this category of cortical regions. How representative is the phenomenon that the authors observed for the wM1-S1 interaction regarding the frontal cortical modulation of sensory processing? It is thus an overreaching conclusion, not strongly supported by the current results, for the following statement in the abstract, “In contrast to current models of frontal cortex function, frontal cortex did not substantially modulate the response amplitude of preferred stimuli. Rather, frontal cortex specifically suppressed the propagation of distractor stimulus responses, thereby preventing target-preferring neurons from being activated by distractor stimuli.”

Thank you for your comments. We agree with your assessment that the wMC-S1 sensorimotor system may be uniquely adapted for active whisker sensing, and therefore the more general label of ‘frontal cortex’ is potentially misleading. For this reason, we now refer to our findings as specifically relating to motor cortex, including in the Title and Abstract. Also, we have rewritten the Introduction to focus on the influences of motor cortices on sensory processing, within the frameworks of (for example) corollary discharge, spatial attention, and active sensing.

Similarly, in line 263, the authors stated, “From these data, we conclude that the main impact of wMC on sensory encoding is to suppress the propagation of distractor stimulus responses in to target-aligned S1.” There are certainly other important, and probably also ‘main’, impacts of M1 on sensory processing, e.g., as were demonstrated in Xu et al, 2012, Ranganathan et al, 2018. The phenomenon observed in the current study is not necessarily the main effect of M1 on S1.

Our use of 'main impact' was in reference to our dataset and was not meant to exclude other studies of these circuits. We have now taken out any references to 'main impacts' or other such general conclusions.

2. A major drawback of this study is the lack of recording of wM1 activity and the related whisking motion. It has been established that wM1 neurons are driven by both whisker sensory input and active movement of whiskers (Huber et al, 2012), and the activity in the wM1-S1 projections has been shown to encode whisker motion information (Petreanu et al, 2012). Moreover, the whisker system is an active sensing system, where sensory processing involves active movement of the sensors. Whisker stimulation on the two sides differentially defined as target and distractor and respectively associated with go and no go responses would certainly evoke distinct whisker movement patterns. Such differential whisker movements are likely to play an important role in the discrimination of target vs. distractor stimulus, and could be associated with different activity patterns in the wM1 in the two hemispheres. To understand how wM1 contributes to the sensory dependent go/no go behavior, it is important to monitor the whisker motion and record neuronal activity in wM1 during the task. Otherwise, some of the major results in this study stays at phenomenological level, and remain difficult to understand. For example, why suppression of wM1 in either hemisphere both led to increased false alarm rate (Figure 1)? How did the optogenetic suppression of wM1 affect whisking motion? Why suppression of wM1 contralateral to target stimulus marginally changed the activity of ipsilateral S1 while selectively increased the d-prime for the ipsilateral distractor (Figure 3)? Why suppression of wM1 contralateral to distractor did not have any effect? Could such asymmetrical effects be due to asymmetry in whisker motion and in wM1 activity between the two hemispheres?

Thank you for identifying these potential confounds. First, we describe previous analyses that may inform these questions; second, we describe new experiments and analyses.

We begin with a discussion about activity in wMC. We have extensive experience recording spiking activity in wMC during expert performance in this task. Target-aligned wMC recordings were published in Zareian et al., 2021 eNeuro; both target-aligned and distractor-aligned wMC recordings are posted in Zareian et al., 2022 bioRxiv. We observed lateralized sensory responses in wMC (larger target-evoked responses in target-aligned wMC and larger distractor-evoked responses in distractor-aligned wMC; Zareian et al., 2022 bioRxiv Fig. 4) as well as attenuation within wMC (larger target-evoked responses in target-aligned wMC compared to distractor-evoked responses in distractor-aligned wMC, Zareian et al., 2022 bioRxiv Fig. 4 E, F). However, we do not believe that wMC sensory-evoked responses can account for the differences in S1 propagation observed in this study. S1 stimulus encoding of the preferred stimuli (target stimuli in target-aligned S1 and distractor stimuli in distractor-aligned S1) is overlapping for the first 40 ms post-stimulus (Fig. 2 E, middle). Across this same early time window there is robust divergence of the cross-hemispheric propagation of these stimulus responses (Fig. 2 G, middle). This early divergence was the primary motivation for considering a proactive wMC-S1 modulation (discussed further in the next comment).

Shown below are comparisons of S1 and wMC recordings of their preferred stimuli in expert mice (target-aligned responses to target stimuli are in purple and distractor-aligned responses to distractor stimuli are in green). wMC recordings are from Zareian et al., 2022 bioRxiv. Note that in both structures the divergence occurs after 40 ms post-stimulus. This is well after the divergence of cross-hemispheric propagation of target and distractor stimuli in S1. (Shown in [B] are recordings from deep layers of wMC. Spiking activity from superficial layers did not show a target-distractor divergence.)

Regarding whisker movements, we emphasize that our task is a passive sensing task and therefore whisker movements play a fundamentally different role than in the active sensing tasks of Huber et al., 2012 and Petreanu et al., 2012. For example, we and others have demonstrated that passive whisker stimulus detection is enhanced when not actively moving/whisking (Ollerenshaw et al., 2012; Kyriakatos et al., 2017; Marrero et al., 2022). However, we certainly agree that whisker movements (evoked by sensory or optogenetic stimulation) can have massive impacts on S1 activity. Therefore, we performed a series of new whisker monitoring experiments and analyses (see below). While these new data do not change our main conclusions, we agree that our study would have been incomplete without them.

Why suppression of wM1 in either hemisphere both led to increased false alarm rate (Figure 1)? In response to this question, we considered whether the similar behavioral effects may be explained by the robust inter-hemispheric interactions between bilateral wMCs (i.e., are we effectively suppressing both wMC regions when directly illuminating one region?). (As also mentioned to Reviewer 1) we experimentally addressed the specific question: what percentage of spiking in target-aligned wMC is suppressed by optogenetic inhibition of target-aligned wMC (direct) compared to optogenetic inhibition of distractor-aligned wMC (indirect)? As shown in Supplementary Fig. 1, spike rate suppression of putative excitatory neurons in target-aligned wMC was 74% (direct) and 36% (indirect). Thus, the similar behavioral effects from suppressing either hemisphere may be due to the robust coordination between wMC regions. These new findings and interpretations are described in the Results.

How did the optogenetic suppression of wM1 affect whisking motion? Why suppression of wM1 contralateral to target stimulus marginally changed the activity of ipsilateral S1 while selectively increased the d-prime for the ipsilateral distractor (Figure 3)? Why suppression of wM1 contralateral to distractor did not have any effect? Could such asymmetrical effects be due to asymmetry in whisker motion and in wM1 activity between the two hemispheres? To address these questions, we conducted bilateral whisker imaging experiments during expert task performance, with and without optogenetic perturbation. Our transient wMC optogenetic suppression did not consistently impact whisker movements (as determined by whisker motion energy). This was verified both before whisker stimulus onset (Supplementary Fig. 9) and during the early post-stimulus window (Fig. 3 B,C,E,F, bottom rows). Importantly, increased distractor stimulus propagation into target-aligned S1 with wMC suppression cannot be accounted for by changes in whisker motion energy (Fig. 3 C, bottom row).

As you suggest, we did observe differences in whisker motion depending on stimulus identity, with greater increases in whisker motion energy following target stimuli and initiating earlier in target-aligned whiskers (Fig. 2 E,G, bottom rows). From these analyses we now recognize that the persistent target-evoked activity in target-aligned S1 (Fig. 2 E, middle) may be a consequence of whisking (Fig. 2 E, bottom). In contrast, the early, robust propagation of target-evoked activity into distractor-aligned S1 (Fig. 2 G, middle) does not appear to be driven by whisker movement, since robust movement increases (Fig. 2 G, bottom) lag the neuronal propagation. These critical new findings are described in the Results. And yet, we do recognize that we cannot entirely rule out contributions from small changes in whisker movements below our imaging resolution, which we mention in the Discussion.

3. The claim of proactive effect of wM1 on S1 is preliminary. It is certainly an important concept that some of the top-down inputs exert proactive or predictive effect. In the current study, the authors showed that the light stimulation before the whisker stimulus onset also had an effect on S1 pre-stimulus activity. However, the light stimulation covered both the pre-stimulus and post-stimulus periods. The authors did not examine whether the pre-stimulus suppression of wM1 only did indeed impact the encoding of either target or distractor stimulus in S1, or the behavioral performance. To claim that the wM1 proactively modulate sensory processing in S1, the authors should restrict the suppressing light stimulation of wM1 within the pre-stimulus time epoch, and examine whether there would be a similar effect on behavior as shown in Figure 1G, or on sensory coding as shown in Figure 3 and 4. Otherwise, the current results cannot distinguish the 'proactive' vs 'reactive' natures of the wM1-S1 modulation, and the last section and Figure 5 are not adding much value to this study.

Our argument about proactive modulation was originally motivated by the extremely rapid divergence of the cross-hemispheric propagation, which is highly statistically significant by 20 ms post-stimulus (Fig. 2 G, middle). Reactive feedback modulation typically emerges ~100 ms post-stimulus, including for the wMC-S1 pathway (Manita et al., 2015). This early difference in S1 cross-hemispheric propagation indicated to us that the sensory filters are in place before stimulus onset, to influence the feedforward

sensory sweep. We recognize that these motivations were not clear in the original manuscript, and we have emphasized them in the revision.

However, we also recognize that these arguments require experimental support. We agree that our data showing wMC pre-stimulus modulation of S1 activity (Fig. 5) is necessary but not sufficient evidence. Therefore, we conducted the experiment you suggested, restricting optogenetic wMC suppression to short epochs around the time of stimulus onset (Supplementary Fig. 13). While all three peri-stimulus suppression windows significantly increased false alarm rates, the only statistically significant increase in distractor detection was from the window peaking at the time of stimulus onset (Supplementary Fig. 13 E, bottom). As a control for opto-stimulation artifacts, we also demonstrate a lack of these effects (increases in false alarm rates and/or distractor detection) in wild type littermates lacking ChR2 expression (Supplementary Fig. 14). We interpret these findings as additional evidence of proactive wMC modulation, setting the initial conditions for the routing of feedforward signals. Yet, we recognize that these findings do not entirely rule out rapid reactive feedback signals, occurring within the first 100 ms post-stimulus. We describe these new findings in the Results and our interpretations in the last paragraph of the Discussion.

Reviewer #3 (Remarks to the Author):

Introduction: The manuscript by Morishita and colleagues uses a combined approach of optogenetics, electrophysiology, and fiber photometry, with cognitive testing. Mice were assessed on a head-fixed go/no-go task, where they were trained to selectively respond to whisker stimulation in one area and withhold responding in another. Optogenetic suppression of the whisker region of motor cortex (wMC) via GABAergic activation was assessed. Electrophysiological recordings of primary somatosensory cortex (S1) were assessed during normal task performance and in combination with wMC suppression. The authors conclude that wMC modulates S1 to primarily facilitate the processing and response inhibition towards non-target stimuli. Additionally, these data indicate towards a top-down modulation of S1 activity by wMC input, prior to the presentation of a stimulus.

The authors use a number of techniques, for the most part well-chosen, that offer a degree of temporal and cellular specificity. In particular, the use of in vivo optogenetics allows within-subjects analysis and causal manipulations, which the authors write have not been used previously to answer this particular question. Multiple controls are provided for the potential confounds of optogenetic stimulation. The paper provides a well-described assessment of the somatosensory cortex during this behavioral paradigm, which could be useful to researchers interested in these processes.

One limitation of the study is the specificity of the manipulations to the circuit under study, and the relationship with behavioral changes observed. The aim is to understand how wMC might modulate S1 and how that modulation affects behavior. However, the optogenetic suppression of wMC will suppress a number of circuits involving wMC – thalamus, striatum, and brainstem for example – and the suppression of any of these circuits could be the cause of any behavioral changes observed. Therefore, the design

of the experiment is not appropriate to determine the causal relationship between wMC suppression-induced S1 changes and behavior. The authors are aware of this, and indeed discuss how some behavioral changes are likely not due to changes in S1. However, it is not difficult, using optogenetics, to isolate the specific projections from wMC to S1; this is a strength of optogenetic approaches as opposed to, e.g., temporary inactivations produced pharmacologically. For me, the lack of specificity is a limitation that could be easily overcome and is necessary to draw the conclusions the authors wish to make.

We agree that this is an important limitation of our study – that the behavioral changes observed from wMC suppression are unlikely to be entirely mediated by changes in S1 activity. However, we disagree that this limitation can be easily overcome with optogenetic terminal suppression. First, wMC modulation of S1 could be through the direct feedback pathway or indirect pathways such as through the thalamus, in the target-aligned hemisphere, the distractor-aligned hemisphere, or both. Given these diverse pathways, identifying the specific pathways involved is a major undertaking on its own. Second, achieving consistent and effective terminal suppression is notoriously difficult (e.g., Mahn et al., 2016). We are actively developing the tools to conduct the proposed experiments but are at the stage of calibrating the effectiveness of terminal suppression for our pathways of interest.

However, to address your general concern, we conducted additional data analyses to better correlate our behavioral and neuronal measures. First, leveraging our catch trials (based on your suggestions below), we now decompose changes in false alarm rates into changes in distractor detection (d') and tendency to respond (criterion) (shown for all conditions in Supplementary Fig. 3). We note two related insights. First, wMC suppression caused an average increase in distractor detection (behavioral d' for distractor stimuli) of 0.22 ± 0.08 , which precisely matches the average increase in target-aligned S1 distractor encoding (neuronal d') in of 0.24 ± 0.05 . Second, in a session-by-session analysis, increases in distractor neuronal encoding in target-aligned S1 has a positive trending correlation with increases in distractor detection ($R^2=0.12$) compared to a near zero correlation with changes in the criterion ($R^2=0.0035$) (Supplementary Fig. 8). These analyses allow for a more nuanced interpretation of our findings: that wMC suppression of distractor propagation into target-aligned S1 likely contributes specifically to reducing distractor-evoked behavioral responses. These analyses and interpretations are included in the Results and Discussion, respectively.

Another question I have concerns the suppression of wMC by stimulating GABA-ergic interneurons. As far as I can tell, there was no electrophysiology done in wMC and therefore no assessment of to what extent this approach does suppress activity in (the pyramidal cells) of wMC. This is important, as there are reports that optogenetic stimulation of frontal cortical interneurons can in fact enhance cognitive function (and in tasks similar to that used by the authors).

Studies that we are most familiar with to enhance cognitive function stimulate frontal cortex PV neurons phasically, to enhance temporal synchronization at specific frequencies. In contrast, our ramping stimulation is designed to achieve a stable

suppression of excitatory neurons. To verify our suppression method, we conducted new recordings of wMC activity during task performance in response to optogenetic wMC suppression (Supplementary Fig. 1). We find that we achieve on average 74% spike rate reduction of putative excitatory neurons with our suppression method.

A third limitation is that the animals under study are likely not normal. The head-fixed procedure is stressful and can drastically raise the level of stress hormones and alter behavior (see e.g., Juczewski et al., 2020, Scientific Reports). Studies using the head-fix method must at least clearly acknowledge this limitation.

We agree that head-fixation is non-ethological and may introduce confounds of stress as well as sensory-motor prediction errors (e.g., Keller et al., 2012). We now mention this in the Discussion, and the need to verify these findings in freely moving behaviors.

I am also unclear what the authors are intending to study psychologically. On one hand, it seems that they wish to study attention: they refer to “detection” of the stimulus, and “distractors”. However, to properly study stimulus detection, the detectability of the stimulus must be manipulated parametrically, and a relationship demonstrated between the detectability of the stimulus and the experimental manipulations by, for example, altering the stimulus duration or intensity of the stimulus. Distractors can be introduced by presenting distracting stimuli simultaneously with the target stimulus. Good examples of how to properly study attention are studies involving the 5-choice serial reaction time task, or the rodent continuous performance task. The set-up the authors use is probably best described as studying the well-learned performance of a simple go/no go task. In this context is the incorrect stimulus best described as a distractor, or is it simply a non-rewarded stimulus? This is important, as it changes the interpretation of the manipulations and the (possibly, see above) behavior-related activity in S1. Furthermore, depending on what they intend to study, alternative tasks – for example a task with symmetrical responses to control for movement confounds, addressed by the authors in the Discussion – might have been more appropriate.

The process that we are intending to study is the ability to filter out non-relevant stimuli, as proposed in the Treisman attenuation model of selective attention. Multiple studies have observed attenuation of non-relevant stimuli along the cortical hierarchy (Moran and Desimone, 1985; Tootell et al., 1998; Treue, 2001; Aruljothi et al., 2020). However, the mechanisms that implement attenuation, thereby preventing responses to non-relevant stimuli, are unknown and are the focus of this study. To study attenuation, we needed a task in which one stimulus is ignored and another stimulus is passed along to higher order processing (in our case, response licking). Therefore, we believe that the simple Go/No-Go task used here is more appropriate than a 2AFC task with symmetric responses for each stimulus.

The original attenuation model referred to ‘attended and unattended’ stimuli or ‘selected and irrelevant’ stimuli (Treisman 1964). ‘Target and non-target’ and ‘target and distractor’ have also been used in the literature to refer to the same concepts. To minimize confusion, we now better define what we mean by ‘distractor’ when describing

the task in the Introduction. In addition to being unrewarded, responding to distractor stimuli is punished with a time-out (resetting the long inter-trial interval).

The control animals may also be a limitation. It is stated only that they are wild type mice. Ideally, they would be identical to the experimental animals but without the expression of ChR2. Furthermore, they must be littermate controls (many journals now require this). Additionally, the authors state that control studies in mice without optogenetic ChR2 expression were performed identically to the ChR2 expressing mice. Data from these mice do not appear to be included in Figure 1, where the optogenetic experiments are described. The authors mention electrophysiological recordings were taken from 2 wild-type mice (page 30), were these the only controls included for the optogenetic manipulations? The sex of the experimental mice was also not included, only that they were bred on site. What was the ratio of male to female mice? Were sex differences analyzed? Again, consideration of sex is rapidly becoming necessary for publication.

Thank you for these important considerations. We have now collected new data from VGAT-ChR2 expressing mice (30 sessions) and non-ChR2 expressing (wild-type) littermate controls (16 sessions). We now include the littermate, wild-type data in Fig. 1 (Fig. 1 I), which demonstrates lack of behavioral effects from optical stimulation alone. Additionally, for all other experiments conducted during the revision (whisker movement imaging, varying suppression windows, recordings in naïve mice), the non-ChR2 expressing controls were littermates of the experimental VGAT-ChR2 transgenic mice.

Our studies were not originally powered to study sex differences. Nonetheless, we compared our main behavioral and physiological outcome measures according to sex, and did not find significant differences. We did not observe significant sex differences in wMC suppression changes in behavioral measures of false alarm rates (target-aligned, $p=0.91$; distractor-aligned, $p=0.55$) or distractor detection behavioral d' (target-aligned, $p=0.16$; distractor-aligned, $p=0.97$). Additionally, we did not observe significant sex differences in wMC suppression changes in physiological measures of preferred stimulus encoding (neuronal d' , target-aligned, $p=0.82$; distractor-aligned, $p=0.31$) or non-preferred stimulus encoding (neuronal d' , target-aligned, $p=0.06$; distractor-aligned, $p=0.52$). The full list of the sex of each mouse and identification of littermates is now included in Supplementary Table 1.

General comments:

- The introduction focuses on describing frontal cortex and its communication with sensory cortices to modulate response inhibition when faced with distracting (non-rewarded? See above) information. The authors then specify that they will be manipulating motor cortex and its pathway with the somatosensory cortex. Was this region chosen as a primary top down modulator of behavioral inhibition, or more so because of the nature of this task specifically? The role of this region specifically in the context of controlling response inhibition to distracting information is not discussed in the introduction.

We now include this important information in the Introduction and Discussion. wMC was chosen due to its robust connectivity with S1, through both direct and indirect pathways. wMC is generally considered to be a 'motor' structure and has not been widely appreciated as a modulator of sensory processing. Therefore, in the Introduction (which we have substantially revised) we now indicate several frameworks (such as corollary discharge, spatial attention, and active sensing) that do propose motor cortex modulations of sensory processing.

Motor cortex modulation of sensory processing in the context of sensory selection had not previously been studied, and therefore we weren't sure whether wMC would contribute to target selection or distractor suppression. Given our findings of wMC contributions to distractor suppression, in the Discussion we now compare this finding to the framework of corollary discharge. We speculate that a general function of motor cortex may be to suppress sensory processing, both in the content of movement (corollary discharge) and goal-direction (in the context of sensory selection). Obviously, this speculation requires additional testing, including in freely moving behaviors.

- In the discussion, the authors focus on the involvement of these regions in distractor suppression, while some features of the data do not suggest this type of selectivity:

- o 1. The data indicate that wMC suppression may increase both hit rate and false alarms, as well as premature responding. This is muddled by the discussion of small and large amplitude stimulation, as well as "depending on the statistical calculations used". The inclusion of these different amplitude stimulations appears under-realized, and it is unclear if there are any considerable differences resulting from using these two different protocols. The authors state that "unless otherwise stated, the data is reflecting large amplitude stimulation", so the purpose of including the small amplitude stimulation could be described more clearly.

The use of two stimulus amplitudes (always 4x different) is to ensure that we are operating within the dynamic psychometric range of detection (with lower hit rates for 'small' compared to 'large' stimuli). This description is now included in the Methods.

We agree that the language 'depending on statistical calculations used' is not helpful. Originally, this was referring to paired t-tests of the raw data versus one-sample t-tests of modulation indices. For simplicity, we now only report the paired t-test analyses since this is the least processed statistical method (requiring no data normalization or transformation).

- o 2. Suppression of wMC led to increases in target-aligned S1 neurons to both target and distractor stimuli.

(Answered together with the next point.)

- o 3. Together, and the authors do mention this, the data here indicate that this circuit may be more specific to the gating of incoming stimuli and associated behavioral responses, whereas wMC suppression leads to more "liberal" assessment of stimuli and responding. However, the first part of the discussion is focused on the distractor/non-

target side of this, and perhaps should be re-structured to reflect that these data do not appear specific to non-target information processing.

We agree with your assessment, and therefore no longer conclude that the effects of wMC are specific to distractor stimuli. Additionally, in the Results we have changed the 2nd and 4th sub-headings to refer to general changes in (behavioral) responding and (neuronal) whisker stimulus encoding from wMC suppression, respectively. As for the bulk of the manuscript, we do focus on distractor detection and distractor encoding because, to us, this is the clearest mechanistic link between behavioral and neuronal measures in our dataset: increased distractor detection due to increased propagation of distractor stimuli into target-aligned S1.

Comments on the Methods section:

- There are a few questions about the task that I am hoping the authors could expand upon:

o Definition of expert: Is this term just being used as a description in this manuscript or is there a basis that having a d' over 1 is considered expert performance. Is 1 close to the upper range of scores that is possible for d' on this task? Ideally a statistical definition of expert-level performance would be given.

For psychology studies, d' effect sizes of approximately 1 have been used to denote 'large' effects (specifically 0.8 in Cohen 1988, Statistical Power Analysis for the Behavioral Sciences p. 80). However, we certainly agree (as does Cohen) that this is highly subjective. The theoretical upper limit for our behavioral calculation is $d'=5.2$ (considering 100 target and 100 distractor trials), and expert mice in our task typically perform around $d'=2$. Perhaps most telling, in our experience, once mice achieve $d'=1$ for three consecutive days, their performance tends to stay above $d'=1$ on subsequent training/testing days, indicating that the mice have indeed learned target-distractor discrimination. While we agree that a statistical definition would be ideal, we are not aware of any such approach that is generally accepted for these types of tasks.

o Session criteria: What led to the termination of a session, if both the number of trials (200-400) and the time (1-2 hours) were variable? Were all animals subjected to an equal number of trials, or a minimum, for their data to be compared to each other?

All mice were allowed to continue in the task until unmotivated, defined as >2 min of no licking and >3 min of no 'hit' trials. Depending on mouse size, training stage, reward history, and likely other factors, a full daily session could vary from 200-400 trials occurring over 1-2 hours. In post-processing of this raw session data, we set strict criteria for deciding which segment to use for further analyses. We use the standardized criteria of a single continuous period of task performance greater than 10 minutes without a pause in responding (licking) greater than 60 sec. For each session, only one engaged period (the longest one) is used for subsequent analyses. These criteria are now better described in the Methods. This standardized inclusion criteria ensures that we are comparing periods of high engagement between sessions and mice. A session

is not used for any analyses if there is no single continuous period of engagement of at least 10 minutes.

o Task disengagement trials: It is mentioned that animals only needed to complete a minimum of 5 trials from either engagement type to be considered for analysis. Could the authors include a mention of how much variance there was across their cohort?

We now include these descriptive statistics for all trial types in Supplementary Table 2.

o Non-rewarded catch trials: On page 5, the authors state “catch trials (spontaneous response rate) when describing their measures. Could the authors please briefly elaborate why the catch trials are included and the purpose they uniquely serve?

Thank you for pointing out this lapse in our analyses. We hadn't previously reported our catch trial data. Upon reanalysis, we recognized that wMC suppression increased catch rates as well as false alarm rates, which we now report (Fig. 1 G, middle row). These data informed us that part of the increase in false alarm rates may be due to a general increase in the tendency to respond. Therefore, we now use signal detection theory to convert changes in false alarm and catch rates to changes in detection (d') and tendency to respond (c) (Fig. 1 G, bottom row, and Supplementary Fig. 3). Furthermore, these analyses led to our realization that the increased propagation of distractor stimuli into target-aligned S1 likely accounts for the specific changes in distractor detection (Supplementary Fig. 8). These new analyses and interpretations are included in the Results and Discussion.

o Analysis for response measures = 0: “Response rates of 0 and 1 were adjusted as 0.01 and 0.99 respectively to avoid yielding infinite values.”

Can the authors discuss why this correction method was used, or provide reference if this practice has been published before? Other methods correct for zero misses/false alarms using a formula to calculate a d' that take into account the potential possibility of one of these response types occurring in the future. For reference:
<https://link.springer.com/article/10.3758/BF03203619>

Thank you for this reference, and for a statistically rigorous method to bound our d' analyses. We now use the ‘log-linear rule’ as described in the referenced paper throughout (for all behavioral d' data in this study) and include its description in the Methods.

- The masking used to reduce confounds of the optogenetics resulted in noise in the 10hz and 20hz range. Can more details be included to describe the filtering that was done to remove this noise? Could removing neural activity in these ranges from the analysis be missing true neural activity, as in do the neurons being recorded from tend to fire in this frequency during normal task epochs?

The 10 Hz and 20 Hz noise was due to the 10 Hz flicker of the visual mask. To verify that our filtering isn't obscuring our electrophysiological analyses, we present the key

analyses of Fig. 3 without these filters (presented in Supplementary Fig. 7). The main physiological findings (in particular, increased distractor propagation to target-aligned S1 during wMC suppression) are evident with or without filtering. Moreover, we have added additional details of spike filtering in the Methods.

- When describing the calcium imaging data, the following is included, “The imaging datasets analyzed here were previously published (Aruljothi et al., 2020; Marrero et al., 2022). The dataset consists of n=40 behavioral imaging sessions from n=5 mice. The analyses presented in this study are from the 100-200 ms post-stimulus imaging frame, during the lockout period and before the response window.”

o Can the authors confirm this specific analysis was not included in previous publications, or mention what is novel about the data as it is shown in this manuscript?

The imaging data was collected and analyzed previously in (Aruljothi et al., 2020; Marrero et al., 2022). However, we did not statistically compare or report the cross-hemispheric propagation of target and distractor stimuli. Thus, the reported analyses and conclusions are novel.

Comments on the Result section:

When discussing the wMC suppression on pages 5 and 6:

- “Suppression of target-aligned or distractor-aligned wMC caused trends towards increased small amplitude hit rates (with statistical significance depending on calculation method, Supplementary Fig. 1A,B).”

o Can the authors specify what statistics were used to assess the optogenetic effects and were they consistent for all measures?

As mentioned above, we agree that the language ‘depending on statistical calculations used’ is not helpful. We now only report the paired t-test analyses since this is the least processed statistical method (i.e., without additional data normalization/transformation) (Fig. 1 and Supplementary Fig. 2).

- S1 suppression: Could a c (response bias – signal detection theory calculated as $-0.5(z(HR) + z(FAR))$) measure be used with these data to characterize if the bi-directional effects of wMC and S1 suppression are generalized reduction or increases in response rates? (Rather than specific to hit rate or false alarm rates statistically.)

Excellent suggestion. We now include signal detection theory measures of d' and c in our behavioral analyses. As shown in Supplementary Fig. 3, we now plot changes in c versus changes in d' for each experimental condition. As you suggest, target-aligned S1 suppression consistently resulted in increases in c (reduced tendency to respond, green circles, y-axes) whereas wMC suppression consistently resulted in decreases in c (increased tendency to respond, magenta and orange circles). For distractor whisker stimuli, wMC suppression additionally increased distractor detection. (Supplementary Fig. 3 A and C). More generally, with this method we find that most neuronal perturbations resulted in combined effects on both detection and the tendency to respond.

- Modulation index characterization Figure 1G: The results for the modulation depict responses across all sessions and appear to include the combination of large vs. small amplitude trials.

o Can the authors confirm if this is the case, and if so, why these data were combined? Was statistical analysis included to confirm the small vs. large amplitude stimulation were not significantly different?

We agree that we should not have combined data from large and small amplitude stimuli, and now only report each condition separately (large amplitude stimuli in Fig. 1 and small amplitude stimuli in Supplementary Fig. 2).

- Figure 1 legend: "Optogenetic suppression (blue light-on) of wMC or S1 was performed on 33% of trials, randomly interleaved with control (light-off) trials."

o Can the authors elaborate why optogenetic stimulation occurred at this rate, as opposed to an even split of trials with optogenetic stimulation vs. without?

Our selection of optogenetic trials is a balance of two factors. We would prefer to use low rates, so that mice do not change their behavioral strategy in response to the optogenetic perturbation. However, using higher rates (up to 50% of optogenetic trials) provides greater the statistical power. We use 33% of trials as a compromise between these two interests. In our experience, and under our specific conditions, our protocol does not cause sustained behavioral changes that impair interleaved control (non-opto) trials. For reference, trial numbers for optogenetic suppression and control conditions are now included in Supplementary Table 2.

When discussing electrophysiological recordings of S1 neurons on pages 8-10:

- Figure 2 legend: There is unclarity in the use of the term "d prime" as an index of neuronal activity due to the use of d prime as a primary behavior measure in the cognitive task. Similarly, using the term "stimulus" to describe both whisker stimulation and optogenetic stimulation. For both of these, it may be better to choose difference words to avoid confusion.

o It may be worthwhile to create a variation of this name or include the definition in the main text (as opposed to only in the figure legend).

Agreed. We now always refer to 'neuronal d prime' and 'behavioral d prime', to distinguish between each type of analysis. Furthermore, throughout the manuscript we refer to 'whisker stimulation' and 'optogenetic suppression' (or 'opto-stimulation', particularly for wild type mice in which we are not driving inhibition).

o It may be ideal to keep the axes in d prime graphs E and G consistent (max 1.6 vs. max 1.2, respectively).

Agreed, done.

- “Neuronal activity during task disengagement was obtained from the same expert mice, after they stopped responding within a session (presumably due to satiety).”
o Could task disengagement result from a reduction in attention, opposed to simply motivation? Since the trial counts/time weren’t the same for each animal, did some animals show prolonged engagement, or did engagement tend to fall off at similar time points during a session?

Engagement times vary widely according to a range of factors, including body weight and hydration status. Based on our understanding of attention, we believe that reductions in attention would manifest as reduced correct responding, without necessarily changes in overall rates of responding. Since our criteria for disengagement are based on cessation of all responding, we believe that this is most likely reflecting a more global change such as motivation and/or satiety.

- “We found that target stimuli induced significant activation of distractor aligned S1 (n=40 sessions, d prime 0.14 ± 0.03 , one sample t-test $p=0.0002$). In contrast, distractor stimuli induced significant suppression of target-aligned S1 (n=40 sessions, d prime -0.07 ± 0.02 , one sample t-test $p=0.007$; paired t-test comparing target and distractor propagation, $p=1.8 \times 10^{-5}$ 207).”

o Can the authors confirm if the wording in the first sentence of this statement is correct? When looking at Figure S2, it appears that target stimuli induced activation of target aligned S1, not distractor aligned S1.

Confirmed. While target stimuli do indeed strongly activate target-aligned S1, these analyses focus on the propagation of target and distractor stimuli across hemispheres (target stimuli into distractor-aligned S1 and distractor stimuli into target-aligned S1). Note, the analyses reflect the data in the ROIs indicated by the red boxes.

- “In marked contrast, disengagement led to an increase in distractor encoding in target-aligned S1”.

o Could this finding be expanded upon in the discussion? As this was the only condition that increased during task disengagement, are there conclusions to be drawn from this finding? It is briefly mentioned on page 16, with the discussion that task disengagement led to reduced stimulus selectivity.

We also find this observation highly intriguing. We now include in the Discussion the observation that during both anesthesia and task disengagement, target-aligned S1 neurons are less selective (respond more to distractor stimuli) than during expert task performance. These findings suggest active suppression of distractor whisker stimulus propagation with task learning and engagement.

When discussing the effect of wMC suppression on S1 activity on pages 12-13

- “From these data, we conclude that the main impact of wMC on sensory encoding is to suppress the propagation of distractor stimulus responses into target-aligned S1.”

o Could the authors expand upon this conclusion? From the data, it appears that wMC suppression generally increased activity in target-aligned neurons regardless of stimulus type, as evident from increases in encoding target and distractor stimuli. Is this conclusion related to the differences in the time-windows that these populations showed increased activity?

Based on this comment and another comment above, we no longer conclude that the effects of wMC are specific to distractor stimuli.

When discussing the context-dependent nature of wMC modulation of S1
- “For recording conditions of anesthetized, awake behaving target-aligned, and awake behaving distractor-aligned, wMC robustly drives pre-stimulus spiking in S1 neurons.”
o Could the authors expand on the animal’s perception of the stimulus under anesthesia? How can pre-stimulus activity can be an index of top-down control of cognition in animals that are anesthetized? Surely this would indicate some kind of anticipatory period?

We do not believe that the wMC drive of S1 activity during anesthesia is related to cognition. Instead, it may be related to memory consolidation or synaptic homeostasis (e.g., Tononi and Cirelli 2014).

REVIEWERS' COMMENTS

Reviewer #1 (Remarks to the Author):

The authors have performed substantial new experiments and analyses in the revised manuscript, and they have addressed all of my concerns. I only have a few suggestions (without requiring further reviewing) :

1. In the abstract, the abbreviations of 'wMC' and 'S1' are not defined when they first appear (line 24, 25). The 'whisker motor cortex' and 'sensory cortex' in line 27 can be replaced by 'wMC' and 'S1', once these are defined in the earlier sentences.

2. Line 27-28 of the abstract, ' ... proactive top-down modulation from whisker motor cortex to sensory cortex, through the differential activation of putative excitatory and inhibitory neurons'.

It is better to explicitly state that the differential activation occurs in the pre-stimulus period, such as: '...through the differential activation of putative excitatory and inhibitory neurons before stimulus onset'.

3. The supplementary figures were not cited in consecutive numerical order. For instance, before line 284 the supplementary figures were Supplementary Fig. 1 – 6, however, Supplementary Fig. 15 (instead of Supplementary Fig. 7) was cited in line 284 – 285.

Reviewer #2 (Remarks to the Author):

In this revision, the authors provided new experimental data and analyses based on my suggestions, and made extensive textual changes to improve their interpretations. Most of my previous concerns have been addressed.

In addition, I would like to suggest that they add the new data in Supplementary Fig. 13 to one of the main figures. In my view, the behavioral effects from temporally more restricted optogenetic inhibition of wMC are as important as the effects on spike rate of S1 shown in Fig. 5.

Reviewer #3 (Remarks to the Author):

The authors have provided an updated version of this manuscript, including multiple additional experiments and analyses to address previous concerns. The authors have also re-organized the scope of this study in terms of the language to more accurately discuss their experiments and results in the context of the specific pathway they are studying. Overall, the manuscript provides an improved and more effective presentation of this study. However, a lingering concern is the relationship between the casual manipulations and physiological recordings, and how these data work together to address the question at hand. Additionally, the results discussing the optogenetic data require some clarification. These concerns are detailed below and could potentially be addressed with edits to the results and discussion section with substantial revisions.

Major Comments:

Results:

There are concerns with the interpretation of the data describing the optogenetic suppression of wMC. Overall, the results suggest that this manipulation generally increases stimulus responding, and is not specific to target or non-target stimuli. As such, the description of these data needs to be thoroughly adjusted to represent the findings.

- Line 114: "...and increased distractor detection (increase in behavioral d')". Based on the calculations included in the method, the d' measure is one of discrimination sensitivity derived from signal detection theory. As such, this measure accounts for both hit rate and false alarm rate, and therefore is not a sole measure of distractor detection. Instead, a high d' measure indicates that mice discriminate better the target from non-target stimuli (increased d' = better

discrimination). Can the authors confirm if this is accurate, and if so, adjust the wording of their conclusions accordingly.

- Line 117: “.but did not increase target detection (target-aligned wMC, light-off trials: 2.17 ± 0.13 , light-on trials: 1.71 ± 0.11 , paired t-test $p = 3.6 \times 10^{-4}$ ”. The wording here should be more accurate. While the “t dect” measure is not significantly increased, it is reduced, as the p-value is 0.00036 and therefore significant. This significant result is also represented in Figure 1G for target-aligned wMC. Further, wMC suppression did significantly increase hit rate, which is itself a measure of target detection. Therefore, the conclusion on Line 119: “From these analyses we conclude that in expert mice wMC suppresses both the tendency to respond and the detection of distractor stimuli”, appears inaccurate based on these data. The authors need to clarify the wording of these results, as it appears the true effect is simply a general increase in responding.

Discussion:

- The authors include some valid conclusions in relation to the physiology of sensory stimuli processing between these regions. A remaining concern is how the discussion of the behavior (Experiment 1) fits into the overall story. I do appreciate the authors’ mention of additional structures potentially being involved, and the plan to run future experiments to address this possibility, as well as the inclusion of response bias to paint a more complete picture of the behavioral profile. However what is still missing for me is a discussion of how the underlying physiological findings they report jive with the results from the causal manipulations of this pathway. Once such a discussion is in place, then a discussion of future experiments to narrow down the mechanisms can be included and will make more sense in that context. Further, do the opposing effects of wMC and S1 suppression on responding bias not warrant a further discussion? Do these effects align with what is observed at the physiological level, or paint a fuller picture of the relationship between these two structures? I would suggest that a more coherent relationship between the behavior and underlying physiology be proposed for this paper to be acceptable in Nature Communications.

Minor Comments:

Introduction:

- Line 61: Comma need after “Previously”.
- Line 65: There is a bit of confusion here, stating stimulation and suppression studies show variable impacts of wMC on S1 through activation, inhibition, or disinhibition. Is there a general consensus regarding whether activating wMC increases S1 activity, or inhibiting wMC reduces S1 activity? Is it dependent on the technique and/or specificity of stimulation/inhibition? If not, it may be better stated that altering wMC activity in either direction can elicit variable changes to S1 activity. Perhaps rewording can more directly inform the reader as to the relationship between activity in these regions.
- Line 70: “We find evidence for wMC suppression of behavioral responses to distractor stimuli...”. See above comments related to the effects of optogenetic suppression, and how these data do not appear specific to distractor stimuli (Figure 1: increased hit rate with wMC suppression).
- Overall, the introduction section is improved and more informative as to why this circuit is of interest to the question of the study.

Results:

- Line 286: “We recorded S1 neuronal activity described above while also applying interleaved optogenetic wMC suppression” Is Figure 3 displaying calcium imaging data? Many would argue this method provides a proxy of neural activity, rather than neural activity directly. It might also help the reader just to say explicitly that these experiments combined optogenetics with calcium imaging (see comment above).
- Line 293: “wMC suppression transiently increased target stimulus encoding in target-aligned S1 (Fig. 3 B, middle), which we interpret as wMC marginally suppressing target stimulus encoding under control conditions”. It is up to the authors, but the wording here could exclude “control conditions”, and simply say that the interpretation is that the function of wMC is to suppress target stimulus encoding.
- Line 357: “In control (light-off) conditions (Fig. 4 B, left), this neuron responds robustly to target stimuli, but not to distractor stimuli. During wMC suppression (Fig. 4 B, right), responses to target stimuli are maintained, yet now we observe increased responses to distractor stimuli.” Are there statistical analyses to indicate a significant difference between these two conditions for responses to distractor stimuli under wMC suppression? If not this conclusion cannot be drawn.
- Line 477: “For recording conditions of anesthetized, awake behaving target-aligned, and awake

behaving distractor-aligned, wMC robustly drives pre-stimulus spiking in S1 neurons." wMC suppression drives pre-stimulus spiking?

- Line 492: "Both excitatory and inhibitory S1 neurons were significantly driven by wMC in anesthetized and awake behaving mice, both target-aligned and distractor-aligned." Similar to above, but is this activity a result of wMC suppression or is it stimulus driven? Adjusting the wording to mention will aid in the reader's understanding. Additionally, using wording like "S1 neuron activity increased as a result of wMC suppression" helps avoid the question of whether the actual "driver" is the wMC or some downstream/off-target region affected by optogenetic perturbation of wMC.

Reviewer #1 (Remarks to the Author):

The authors have performed substantial new experiments and analyses in the revised manuscript, and they have addressed all of my concerns. I only have a few suggestions (without requiring further reviewing) :

1. In the abstract, the abbreviations of 'wMC' and 'S1' are not defined when they first appear (line 24, 25). The 'whisker motor cortex' and 'sensory cortex' in line 27 can be replaced by 'wMC' and 'S1', once these are defined in the earlier sentences.

Fixed.

2. Line 27-28 of the abstract, ' ... proactive top-down modulation from whisker motor cortex to sensory cortex, through the differential activation of putative excitatory and inhibitory neurons'.

It is better to explicitly state that the differential activation occurs in the pre-stimulus period, such as: '...through the differential activation of putative excitatory and inhibitory neurons before stimulus onset'.

Done.

3. The supplementary figures were not cited in consecutive numerical order. For instance, before line 284 the supplementary figures were Supplementary Fig. 1 – 6, however, Supplementary Fig. 15 (instead of Supplementary Fig. 7) was cited in line 284 – 285.

Fixed.

Reviewer #2 (Remarks to the Author):

In this revision, the authors provided new experimental data and analyses based on my suggestions, and made extensive textual changes to improve their interpretations. Most of my previous concerns have been addressed.

In addition, I would like to suggest that they add the new data in Supplementary Fig. 13 to one of the main figures. In my view, the behavioral effects from temporally more restricted optogenetic inhibition of wMC are as important as the effects on spike rate of S1 shown in Fig. 5.

Done (new Figure 6).

Reviewer #3 (Remarks to the Author):

The authors have provided an updated version of this manuscript, including multiple additional experiments and analyses to address previous concerns. The authors have also re-organized the scope of this study in terms of the language to more accurately discuss their experiments and results in the context of the specific pathway they are studying. Overall, the manuscript provides an improved and more effective presentation of this study. However, a lingering concern is the relationship between the casual manipulations and physiological recordings, and how these data work together to address the question at hand. Additionally, the results discussing the optogenetic data require some clarification. These concerns are detailed below and could potentially be addressed with edits to the results and discussion section with substantial revisions.

Major Comments:

Results:

There are concerns with the interpretation of the data describing the optogenetic suppression of wMC. Overall, the results suggest that this manipulation generally increases stimulus responding, and is not specific to target or non-target stimuli. As such, the description of these data needs to be thoroughly adjusted to represent the findings.

- Line 114: "...and increased distractor detection (increase in behavioral d')". Based on the calculations included in the method, the d' measure is one of discrimination sensitivity derived from signal detection theory. As such, this measure accounts for both hit rate and false alarm rate, and therefore is not a sole measure of distractor detection. Instead, a high d' measure indicates that mice discriminate better the target from non-target stimuli (increased d' = better discrimination). Can the authors confirm if this is accurate, and if so, adjust the wording of their conclusions accordingly.

We confirm that our description in the manuscript is accurate. We applied two different types of behavioral d' analyses in this study: detection d' and discrimination d'. Both d' analyses are calculated based on signal detection theory. (See Methods lines 855-857 for the specific equations.) Of note for this comment and comments below, the 'detection' analyses refer to calculations in which hit rates or false alarm rates are normalized by catch rates (which we refer to as 'target detection' and 'distractor detection', respectively). Thus, the distractor detection d' referred to here is a measurement of behavioral responses to distractor stimuli normalized by the spontaneous (catch) response rate.

- Line 117: "...but did not increase target detection (target-aligned wMC, light-off trials: 2.17 ± 0.13 , light-on trials: 1.71 ± 0.11 , paired t-test $p = 3.6 \times 10^{-4}$ ". The wording here should be more accurate. While the "t dect" measure is not significantly increased, it is reduced, as the p-value is 0.00036 and therefore significant. This significant result is also represented in Figure 1G for target-aligned wMC. Further, wMC suppression did significantly increase hit rate, which is itself a measure of target detection. Therefore, the conclusion on Line 119: "From these analyses we conclude that in expert mice wMC suppresses both the tendency to respond and the detection of distractor stimuli", appears inaccurate based on these data. The authors need to clarify the wording of these results, as it appears the true effect is simply a general increase in responding.

We agree that target detection is significantly decreased. We now indicate that when reporting the data.

Yet, target detection is different from hit rate; for target detection, hit rate is normalized by catch rate to account for general increases in responding. The observation that hit rate is increased and yet target detection is reduced indicates that the increases in hit rate can be fully accounted for by increases in general responding. This was observed for both large (Fig 1) and small (Supplementary Figure 2) amplitude target stimuli.

These findings for target stimuli contrast with our findings for distractor stimuli. With wMC suppression we observed increases in false alarm rate and increases in distractor detection (for both large and small amplitude distractor stimuli). This means that mice responded to distractor stimuli above and beyond what could be accounted for by increases in general responding.

Together, we believe that our findings fully support our conclusion (lines 119-121): “From these analyses we conclude that in expert mice wMC suppresses both the tendency to respond and the detection of distractor stimuli.”

Discussion:

- The authors include some valid conclusions in relation to the physiology of sensory stimuli processing between these regions. A remaining concern is how the discussion of the behavior (Experiment 1) fits into the overall story. I do appreciate the authors' mention of additional structures potentially being involved, and the plan to run future experiments to address this possibility, as well as the inclusion of response bias to paint a more complete picture of the behavioral profile. However what is still missing for me is a discussion of how the underlying physiological findings they report jive with the results from the causal manipulations of this pathway. Once such a discussion is in place, then a discussion of future experiments to narrow down the mechanisms can be included and will make more sense in that context. Further, do the opposing effects of wMC and S1 suppression on responding bias not warrant a further discussion? Do these effects align with what is observed at the physiological level, or paint a fuller picture of the relationship between these two structures? I would suggest that a more coherent relationship between the behavior and underlying physiology be proposed for this paper to be acceptable in Nature Communications.

We believe that paragraph 1 of the Discussion effectively describes the specific relationships revealed by this study between behavior and its underlying physiology. Perhaps the clearest linking sentence is (line 568-569): “This increase in distractor stimulus encoding matches, and likely underlies, the increase in distractor stimulus detection during wMC suppression”. In general, we propose that our findings of increased distractor stimulus propagation into target-aligned S1 (physiological measure) can account for increased distractor detection (behavioral measure).

And yet, we also recognize that there are elements of our behavioral findings that are not clearly reflected in the physiology (lines 629-635). In particular, the general increase in the tendency to respond. For this behavioral outcome we speculate as to other potential neuronal mechanisms.

Minor Comments:

Introduction:

- Line 61: Comma need after “Previously”.

Done.

- Line 65: There is a bit of confusion here, stating stimulation and suppression studies show variable impacts of wMC on S1 through activation, inhibition, or disinhibition. Is there a general consensus regarding whether activating wMC increases S1 activity, or inhibiting wMC reduces S1 activity? Is it dependent on the technique and/or specificity of stimulation/inhibition? If not, it may be better stated that altering wMC activity in either direction can elicit variable changes to S1 activity. Perhaps rewording can more directly inform the reader as to the relationship between activity in these regions.

This is an excellent point, that requires additional investigation. I would speculate that the general belief in the field is that, overall, wMC activity increases spiking in S1. However, it is possible that the increase in spiking is largely driven by activations of inhibitory neurons, and thereby causing a net increase in relative inhibition over excitation. Additionally, the effects of wMC on S1 undoubtedly vary by layer, cortical state, and behavioral context. Moreover, as we show in our study, these effects do change with learning. An entire review article could (and should) be written on this topic.

We strongly prefer our existing wording since it is concise yet specifies which circuit mechanism were investigated within each study.

- Line 70: “We find evidence for wMC suppression of behavioral responses to distractor stimuli...”. See above comments related to the effects of optogenetic suppression, and how these data do not appear specific to distractor stimuli (Figure 1: increased hit rate with wMC suppression).

We hope that our responses to the ‘Major Comments’ sufficiently address this concern.

- Overall, the introduction section is improved and more informative as to why this circuit is of interest to the question of the study.

Thank you.

Results:

- Line 286: “We recorded S1 neuronal activity described above while also applying interleaved optogenetic wMC suppression” Is Figure 3 displaying calcium imaging data? Many would argue this method provides a proxy of neural activity, rather than neural

activity directly. It might also help the reader just to say explicitly that these experiments combined optogenetics with calcium imaging (see comment above).

Figure 3 displays data from electrophysiological single unit recordings (not calcium imaging). Supplementary Figure 6 is the only dataset obtained from calcium imaging. To clarify, when introducing Figure 3 in the Results we now specify: “We recorded S1 spiking activity as described above...”

- Line 293: “wMC suppression transiently increased target stimulus encoding in target-aligned S1 (Fig. 3 B, middle), which we interpret as wMC marginally suppressing target stimulus encoding under control conditions”. It is up to the authors, but the wording here could exclude “control conditions”, and simply say that the interpretation is that the function of wMC is to suppress target stimulus encoding.

Agreed, done.

- Line 357: “In control (light-off) conditions (Fig. 4 B, left), this neuron responds robustly to target stimuli, but not to distractor stimuli. During wMC suppression (Fig. 4 B, right), responses to target stimuli are maintained, yet now we observe increased responses to distractor stimuli.” Are there statistical analyses to indicate a significant difference between these two conditions for responses to distractor stimuli under wMC suppression? If not this conclusion cannot be drawn.

The single unit we present here is an example neuron that was used to motivate the subsequent population analyses (Figure 4C) and was not the basis for any conclusions of this study. However, we do now include that data and statistics for this single unit example. The increase in distractor responses with wMC suppression was not statistically significant but trending at $p=0.14$ (two-tailed t-test). This contrasts with the statistically significant differences observed across the population of S1 neurons (Figure 4F). Our conclusions are exclusively based on the population analyses.

- Line 477: “For recording conditions of anesthetized, awake behaving target-aligned, and awake behaving distractor-aligned, wMC robustly drives pre-stimulus spiking in S1 neurons.” wMC suppression drives pre-stimulus spiking?

We confirm that our original sentence is correct. Since suppressing wMC resulted in reduced spiking in S1 neurons, we infer that under normal conditions wMC drives spiking in S1 neurons.

- Line 492: “Both excitatory and inhibitory S1 neurons were significantly driven by wMC in anesthetized and awake behaving mice, both target-aligned and distractor-aligned.” Similar to above, but is this activity a result of wMC suppression or is it stimulus driven? Adjusting the wording to mention will aid in the reader’s understanding. Additionally, using wording like “S1 neuron activity increased as a result of wMC suppression” helps avoid the question of whether the actual “driver” is the wMC or some downstream/off-target region affected by optogenetic perturbation of wMC.

We confirm that our original sentence is correct. All analyses in this section of the Results were performed pre-whisker-stimulus (line 471-473) and therefore are not stimulus driven. As indicated above, we infer the normal function of wMC in driving spiking in S1 by the observation that suppressing wMC leads to reduced spiking in S1. We strongly believe that this approach is more valuable than stimulating a brain region, which may evoke a response that is outside the normal physiological regime (including through off-target pathways).